# Process-based classification of Mediterranean cyclones using potential vorticity

Yonatan Givon[1], Or Hess[1] Emmanouil Flaounas[2], Jennifer L. Catto[3], Michael Sprenger[4], and Shira Raveh-Rubin[1]

1.      Department of Earth and Planetary Sciences, Weizmann Institute of Science, Rehovot, Israel

2.      Institute of Oceanography, Hellenic Centre for Marine Research, Athens, Greece

3.      Department of Mathematics and Statistics, University of Exeter, Exeter, UK

4.      Institute for Atmospheric and Climate Science, ETH Zurich, Switzerland

*Correspondence to*: Yonatan Givon (yonatan.givon@weizmann.ac.il)

**Abstract.** Mediterranean cyclones govern extreme weather events across the Euro-African basin, affecting the lives of hundreds of millions. Despite many studies addressing Mediterranean cyclones (MCs) in the last decades, their correct simulation and prediction remain a significant challenge to the present day, which may be attributed to the large variability among MCs. Past classifications of MCs are primarily based on geographical and/or seasonal separations, however, here we focus on cyclone genesis and deepening mechanisms. A variety of processes combine to govern MC genesis and evolution, including adiabatic and diabatic processes, topographic influences, land-sea contrasts, and local temperature anomalies. As each process bears a distinct signature on the potential vorticity (PV) field, a PV approach is used to distinguish among different "types" of MCs. Here, a combined cyclone tracking algorithm is used to detect 3190 Mediterranean cyclone tracks in ECMWF ERA5 from 1979-2020. Cyclone-centered, upper-level isentropic PV structures in the peak time of each cyclone track are classified using the Self Organizing Map (SOM). The SOM analysis reveals 9 classes of Mediterranean cyclones, with distinct Rossby wave-breaking patterns as discernible in corresponding PV structures. Though classified by upper-level PV structures, each class shows different contributions of lower-tropospheric PV and flow structures down to the surface. Unique cyclone life cycle characteristics, associated hazards (precipitation, winds, and temperature anomalies), and long-term trends, as well as synoptic, thermal, dynamical, seasonal, and geographical features of each cyclone class, indicate dominant processes in their evolution. Among others, the classification reveals the importance of topographically-induced Rossby wave breaking to the generation of the most extreme Mediterranean cyclones. These results enhance our understanding of MC predictability, by linking the large-scale Rossby wave formations and life cycles to coherent classes of under-predicted cyclone aspects.

## 1. Introduction:

The Mediterranean has long been described as a global cyclogenesis hotspot and one of the most sensitive regions in terms of global warming (Giorgi and Lionello, 2008, Tuel and Eltahir, 2020). This relationship appears intertwined since Mediterranean cyclones (MCs) form the bulk of extreme weather events across the region, which are the manifestation of the warming climate (Zittis et al., 2022, Hochman et al., 2022 a). MCs are the primary source of extreme precipitation, winds, temperatures, and dust events. As such, MCs pose significant threats to society, and challenges to weather and climate predictions (Flaounas et al., 2022). The impact of under-predicted MCs on the heavily populated Mediterranean coast, and further inland, is tremendous. Flood and wind damage may result in the loss of homes and even lives, while better weather predictions may allow for better warnings and preparations to reduce costs significantly. On climate scales, low confidence in the response of MCs to a changing climate may lead to an underestimation of the severity of the impact of global warming on the Mediterranean and to inadequate infrastructure planning. The recent increase in extreme weather events in the Mediterranean such as droughts, wind storms, and hot spells, are all linked to changes in MCs' behavior (Lionello et al., 2008, Dubrovský et al., 2014 Trigo et al., 2000). MCs are expected to decrease in frequency across the northern Mediterranean, along with an expected poleward shift of the Atlantic storm track, though their intensity may increase in some cases as the Mediterranean becomes warmer and tropical-like cyclones emerge (Gaertner et al., 2007, Cavicchia et al., 2014 a and b). To improve the understanding of present and future extreme weather events in the Mediterranean, it is crucial to better understand their mediators – MCs. Changes in MCs' intensity, frequency, duration, propagation, or extent, all have implications for future climate response (Nissen et al., 2014, Reale et al., 2022).

Mediterranean cyclogenesis has been attributed to downstream development, as Atlantic cyclones contribute to the amplification of ridges over the East-Atlantic, that in turn induces troughs over Europe and cyclogenesis in the Mediterranean. Indeed, Atlantic cyclones were detected as precursors of intense MCs (Raveh-Rubin and Flaounas, 2017). In this context, MCs have been often referred to as daughter cyclones (Romem et al., 2007, Ziv et al., 2015). Mediterranean daughter cyclones can appear in various regions relative to their parent cyclone, commonly along the parental frontal system, but also in the warm sector or linked to a separate synoptic system (Saaroni et al., 2017). Nevertheless, MCs differ strongly by season, location, duration, extent, and impact.

One feature known to modify Mediterranean cyclogenesis is topography. Climatological studies of MCs (Alpert et al., 1990, Campins et al., 2011, Lionello et al., 2006) reveal the uneven distribution of MCs across the basin, with primary cyclogenesis hotspots strongly linked to topography. For this reason, most classifications of MCs focus on geographical and seasonal features (Campins et al., 2000, Maheras et al., 2001, Trigo et., al, 2002, Alpert et al., 2004, Hofstätter et al., 2016, Lionello 2016, Flaounas et al., 2022), coining the Genoa cyclones, North-African ("Sharav") lows, and Cyprus lows as the dominant modes of MCs. Genoa cyclones form in the lee of the Alps and dominate the winter season. These lee-cyclones tend to travel east towards the Balkans, Greece, or Turkey. The "Sharav" lows are known to appear mostly during summer and spring and tend to initiate in the lee of the Atlas (Trigo et al., 1999). Cyprus lows also frequently occur in the winter and autumn seasons, providing the majority of precipitation events in the Middle East (Saaroni et al., 2010). Other recognized climatological hotspots for cyclogenesis include the Adriatic, Aegean, and Black Seas, and the Iberian Peninsula. However, even for a given region of cyclogenesis, large case-to-case variability emerges due to the wide variety of processes affecting MCs (Trigo et al., 2002). With baroclinic instability driving cyclogenesis, the Norwegian model (Henry, 1922) and the refined Shapiro-Keyser model (Shapiro and Keyser, 1990) often describe the lifecycle of extratropical cyclones over the open oceans conceptually. Observed discrepancies with theory gave rise to various approaches to extratropical cyclone classification, as reviewed by Catto (2016). However, these classifications are less relevant for the Mediterranean region, where additional mechanisms strongly modulate cyclogenesis and MC evolution.

The driving mechanisms of Alpine cyclogenesis were reviewed recently by Buzzi et al. (2020). The impact of topography is considered most crucial in the first stage (stage A) of lee-cyclogenesis when the surface cold front is physically blocked by topography. The rapid deepening during stage A is attributed to geostrophic adjustment processes necessary to balance the lag of the cold front impinging on the Alps. This phase lasts until sufficient cold-

air mass penetrates the Mediterranean for the surface cold pool of the Rossby wave to propagate across the barrier. Stage B is then characterized by the development of a frontal structure further away from the mountain, and a continuous, yet slower, deepening of the cyclone by the release of potential baroclinic instability from the synoptic-scale reservoir (Mattocks and Bleck, 1986). As illustrated in Givon et al., (2021), the transition between stages A and B is often manifested in upper levels as filaments of PV streamers and cut-offs attributed to cyclonic and anticyclonic

Rossby wave breaking (hereafter, CWB and AWB, Thorncroft et al., 1993). Tsidulko and Alpert (2001) showed that stage A is dominated by the cyclogenetic contribution of the synergy between adiabatic PV advection and topography, while stage B is purely dominated by PV advection. This constructive interaction between PV advection and topography acts to enhance the vertical motions and deepen the surface cyclone.

       Thermal lows demonstrate another mechanism for cyclogenesis, mostly attributed to regional surface temperature

anomalies leading to local instability and convection (Rácz and Smith, 1999, Johnson, 2003). In the Mediterranean basin, these may be further separated into the continental heat lows, mostly found along North Africa as Sharav lows, and heat lows scattered across the Mediterranean. The first is dry and usually does not precipitate, hence is shallower and shorter-lived (Ammar et al., 2014). The latter can be sustained by a wind-induced surface heat exchange mechanism (WISHE) and grow as a "diabatic Rossby wave", which owes its deepening to evaporation-precipitation

feedback, leading to the release of latent heat by condensation (Lin, 1982, Boettcher and Wernli, 2011 and 2013). Kouroutzoglou et al. (2018) showed the importance of WISHE for the generation of explosive cyclones (Kouroutzoglou et al., 2011), another often used sub-class of MCs. Recently, a growing scientific interest is noted in the appearance of "Medicanes": tropical-like MCs that wreak havoc across Europe (Cavicchia et al., 2014a, 2014b, Flaounas et al., 2022). While a conclusive definition of a "Medicane" is still pending, it is described as a warm-core,

axisymmetric MC with low vertical shear of horizontal wind speed. Medicanes are characterized by intense deepening, precipitation, and wind speeds. Numerical simulations of two Medicane cases (Miglietta and Rotunno 2019) suggest that WISHE may function as an important driving mechanism, though this relationship may be case-dependent.

       While the most common MC classification relies on geographical-seasonal features, no current classification of MCs is based on their dominating driving mechanisms (Flaounas et al., 2022). Campins et al., (2000) performed an objective

analysis of SLP patterns using a cyclone-centered approach, however, the analysis was limited to western MCs, and the results mostly converge back to geographical features. Flaounas et al. (2021) analyzed MCs from a PV perspective and showed the alternating dominance of adiabatic and diabatic processes. Flocas (2000) performed a PV analysis of MCs and asserted the dominance of upper-level PV, as opposed to low/mid-level PV, in the evolution of MCs. Portmann (2020) showed, using forecast ensembles, how cyclone prediction is tightly related to the prediction of the

non-linear upper-tropospheric PV cut-off. The large sensitivity of Rossby-wave evolution to simulated topography makes MCs challenging to predict or numerically reproduce (Smith 1984, Mattocks and Bleck, 1986, Tafferner 1990, Tsidulko and Alpert 2001, Buzzy et al., 2020). The deepening rate and motion of the cyclone are major sources of forecast busts, and a better understanding of the PV-topography relationship is called for (Rodwell 2013, Buzzi et al., 2020, Portmann et al., 2021).

Upper-level PV anomalies are a crucial condition for Mediterranean cyclogenesis and play a major role in MC evolution. Furthermore, as an adiabatically conserved quantity, upper-level PV distributions provide a dynamic diagnostic of the atmosphere, indicative of different dynamical and thermodynamical processes (Hoskins et al., 2015). As such, it is natural to think of upper-level PV in terms of a dynamical classification framework.

       Each cyclone-deepening mechanism is expected to show a unique signature in the PV field, providing the ground for

a systematic, dynamics-based classification of MCs from a PV perspective. More specifically, we aim to address the following question(s):

       1. Are there coherent recurring classes of MCs, from a cyclone-centered, upper-level PV perspective?
       2. Are the different MC classes indicative of dominant deepening mechanisms, involving coherent lower-tropospheric PV signatures and/or near-surface anomalies?

3. How do the classes differ in terms of the variability of their life cycles and associated hazards?

To address the above, we analyze 3190 MC tracks using ECMWF reanalysis ERA5 from 1979 to 2020. Upper-level PV near the cyclone is used to classify MCs. We then explore the geographical, seasonal, thermal, and dynamic features of each class to conclude the dominant process driving the cyclones.

This article is structured as follows: The data and methods are outlined in Sect. 2, with a brief review of the cyclone dataset, clustering approach, and cyclone diagnostics. Results are discussed in Sect. 3, including the total mean patterns of MCs (3.1) followed by the classification results (PV patterns, 3.2), cyclone-track features (3.3), surface impact (3.4), and cluster trends and predictability (3.5). An overview of the results is presented in Sect. 4, followed by concluding remarks in Sect. 5. More details on the SOM analysis, cyclone dynamic features (PV streamers, cut-offs, fronts, and warm-conveyor belts), 3D PV analysis, and Medicane distributions, are included in Appendices A-D.

## 2.   Data and Methods:

To explore the large-scale PV patterns associated with the evolution of the different MC "types", a self-organizing map (SOM) analysis is applied to an MC tracks dataset, classifying upper-level PV structures at the peak time of each cyclone track. Various features are then explored between the different classes, to highlight dominant processes.

### 2.1.        Combined Cyclone Tracking algorithm

In this study, we use composite Mediterranean cyclone tracks from the period 1979-2020, produced from hourly atmospheric fields of ERA5 reanalysis (Hersbach et al., 2020). Composite tracks have been produced using a novel approach (Flaounas et al., 2023) that combines the ensemble of outputs from 10 different cyclone tracking methods to retain only the tracks that concentrate higher agreement among the methods and discard the rest. Composite tracks enable larger confidence levels in the detected cyclones and more distinct dynamical life stages (from genesis to decay), compared to single methods which may present different biases. In this study, we choose composite tracks of confidence level 5, denoting the agreement of at least 5 cyclone detection methods upon the presence of the cyclone for at least one time-step in its lifetime. At this confidence level, the seasonal cycle attains a distinct cycle with a minimum in summer and a maximum in spring, compared to confidence levels of 4 or less, suggesting the removal of shallow heat lows that are of lesser interest from a dynamic perspective (Scherrmann et al., 2023). The cyclone's center position and sea-level pressure (SLP) minimum are tracked throughout its lifetime. Only cyclones that enter the Mediterranean (i.e., any part of their track is present east of 5°W) are accounted for, resulting in 3190 cyclones. The peak time of the cyclone is defined based on the minimum SLP, while cyclogenesis (lysis) is defined as the first (last) time-step in each cyclone track. For ~2% of the cyclones, the minimum SLP of the track is located west of 5°W. To avoid classifying cyclones away from the Mediterranean basin, we iteratively seek a new minimum starting at the following time step, until a minimum is found east of 5°W. However, few (~0.1%) cyclones display anomalous west-bound trajectories, originating east of 5°W and peaking over the Atlantic / West Africa. In these cases, the minimum SLP point is considered the last point in the track that lies east of 5W. Consistently, all other atmospheric data are based on ERA5, interpolated to 0.5-degree horizontal resolution and 3-hourly time steps.

### 2.2.        Self-Organizing Map (SOM) classification

The cyclone tracks are classified at the time of their peak intensity (minimum SLP), according to the regional upper-tropospheric isentropic PV. For each track, the cyclone position is taken at the peak time, and PV on isentropic layers 320-340K (vertically averaged with 5-K intervals) is considered in a cyclone-centered domain of 20° east and west,

20° south, and 40° north of the cyclone center. Note the domain is extended poleward to account for the main PV reservoir. This isentropic layer was chosen so that it intersects the dynamical tropopause across all seasons. The 3190 cyclone-centered PV fields are then classified using a self-organizing map (SOM) algorithm. The choice to classify PV values rather than PV anomalies was made to retain the seasonal separation between the clusters, as well as to obtain the actual flow structure as better illustrated directly by PV. The use of SOM classifications is growing popular in the meteorological field (Liu and Weisberg, 2011, Skific and Francis, 2012, Berkovic et al., 2021). The SOM algorithm relies on a neural network that minimizes Euclidean distances within each cluster, while maximizing the dissimilarity between the clusters, enabling the depiction of distinct patterns. The SOM algorithm maps the variance within the dataset onto an array of composite maps, each corresponding to a reoccurring phase of the parameter under consideration. Similar clusters are placed closer together on the resulting map, while substantially different clusters are placed further from their neighbors. The SOM is a powerful climatological tool that neatly separates the continuum of various states occurring in the system. Here we use a single-layer SOM algorithm, provided by MathWorks (https://www.mathworks.com/help/deeplearning/gs/cluster-data-with-a-self-organizingmap.html, last access: 01/02/2023). The SOM errors, defined as mean squared errors within each cluster and averaged over the entire network, suggest 9 as the optimal choice for the number of clusters, in a 3-by-3 hexagonal configuration (Fig. A1 in Appendix A). Adding further clusters reduced the SOM error at an ever-decreasing rate. This setup results in roughly 300 cyclones per cluster. More details regarding the SOM analysis are presented in Appendix A.

### 2.3. Cyclone-track features

Cyclone features are extracted from the cyclone tracks and compared across the different clusters. These features include seasonal and geographical cyclone occurrence distributions, cyclone propagation velocity, and cyclone deepening rate. The cyclone propagation velocity is evaluated following the track of the cyclone center and averaged along the whole track, defined as:

$$u_{cyclone} = \frac{2\pi r}{360°} cos(\varphi) \frac{d\lambda_{cyclone}}{dt} ; v_{cyclone} = \frac{2\pi r}{360°} \frac{d\varphi_{cyclone}}{dt}$$

Where r is Earth's radius, and $\lambda_{cyclone}$, $\varphi_{cyclone}$ the longitude and latitude of the cyclone center at every time-step. The cyclone maximum deepening rate is quantified with Bergeron units (Tsukijihara et al., 2019, Sanders and Gyakum 1980), given at time t as

$$Bergeron_t = \frac{sin(60)}{sin(\phi_t)} \frac{(SLP_{t-12h} - SLP_{t+12h})}{24h}$$

Where $\phi_t$ is the latitude of the cyclone center at time t. Thus, a 24-hour window centered around time t is shifted along each cyclone track, and the maximum Bergeron number captured for each cyclone track is considered. Values greater than 1 Bergeron imply an explosive cyclone, while negative values imply the minimum depth is reached in less than 12 hours.

### 2.4. Surface Impact

The surface impact of each cluster is evaluated based on deviations from the total cyclone-centered composite average of large-scale and convective precipitation rates and 10-m winds, as well as the deviations of 2-m temperature and upper-level PV from the local monthly climatology (i.e., grid-point based and not cyclone-relative). Gridded student's T-test is conducted on a 99% confidence level (alpha=0.01), tightened by multiple-testing correction criteria (Wilks 2016). As suggested by Wilks (2016), false positives may arise from multiple grid-point statistical testing and can be reduced by imposing an appropriate threshold on the individual P-values, which changes by the number of grid points included in the analysis. Specifically, the threshold is set to:

$$\alpha_{walker} = 1 - (1 - \alpha)^{1/N}$$

Where $\alpha = 0.01$ is the desired confidence level (corresponding to 99% significance) and N=9801 is the number of grid points included in the test. The gridded signal is deemed significant only if its P-value is below $\alpha_{walker}$ (after Walker, 1914).

Precipitation rates are evaluated based on 1-hourly accumulated precipitation until the date under evaluation (ERA5). Both large-scale and convective precipitation are provided by the ERA-5 reanalysis product, using output from parametrization schemes (Hersbach et al., 2020). While these results are parametrized and are expected to show sensitivity to model resolution (especially convective precipitation), it does allow drawing conclusions regarding the sign and relative response of precipitation under the different PV classes.

## 2.5. Dynamical features

To better understand the patterns obtained by the composite means, we consider additional dynamical features that are objectively identified individually for each cyclone:

### 2.5.1. PV streamers and cut-offs

The identifications of PV streamers and cut-offs follow the methodology of Sprenger et al. (2017), which was initially developed and applied for ERA-15 and ERA-Interim (see Wernli and Sprenger, 2007) and then adapted to ERA5. Essentially, PV is interpolated onto a stack of isentropic levels (320 - 340 K, in steps of 5 K). On each of these isentropic PV maps, the largest connected region with PV > 2 PVU is taken as the stratospheric high-PV reservoir, and PV cut-offs are identified as connected regions outside the stratosphere with PV > 2 PVU, i.e., as regions that are surrounded by tropospheric PV air (PV < 2 PVU). The ERA5 grid points on the isentropic levels inside such a PV cut-off feature are labeled as 1, and grid points outside as 0. PV streamers are also identified on single isentropic levels. To this aim, on each isentropic level, the 2-PVU contour delimiting the stratospheric PV reservoir is extracted, and then PV streamers are identified based on this contour as narrow, and elongated disturbances. More specifically, for each point along the contour, it is checked whether there exists a corresponding further point along the contour such that (i) the great-circle (baseline) distance between these two anchor points is less than 2000 km, and (ii) the along-contour distance between the two points exceeds the baseline distance by at least a factor 2. A more detailed description of the algorithm, including some examples, can be found in the Supplement of Sprenger et al. (2017).

### 2.5.2. Warm Conveyor Belts (WCBs)

WCBs are identified according to the methodology introduced by Madonna et al. (2014) and applied to 1-hourly ERA-5 data. Forward trajectories are started globally on an equidistant (80 km) grid at 14 equally spaced layers between 1050 and 790 hPa. The 48-hour trajectories must fulfill two criteria to be classified as WCB trajectories: (i) they must ascend by at least 600 hPa within a 48-hour time window, and (ii) the ascent must occur in the vicinity of an (extratropical) cyclone. A detailed description of the algorithm and some refinements compared to Madonna et al. (2014) are presented in Heitmann et al. (2023). The regions where the WCB trajectories ascend, defined as the vertical range 800 - 400 hPa, are labeled as 1 to obtain an Eulerian WCB weather feature.

### 2.5.3. Fronts

Fronts have been identified using a thermal front parameter method, based on Hewson (1998), and detailed in Sansom and Catto (under review). This method applies to the smooth ERA5 fields of 850-hPa wet-bulb potential temperature ($\theta_w$), then potential fronts are identified where the contours of $\nabla^2|\nabla\theta_w|=0$ (see Figure 3 in Hewson 1998). A mask is applied to keep only the contours that meet some threshold of $\nabla|\nabla\theta_w| \cdot \frac{\nabla\theta_w}{\nabla|\theta_w|}$, known as the thermal front parameter, and where the gradient of $\theta_w$ in the adjacent baroclinic zone meets another threshold. Fronts are designated as cold, warm, or quasi-stationary depending on the frontal speed, and the linear contour features are placed on the ERA5 resolution.

## 3. Results

### 3.1. Mean cyclone climatology

We first examine the whole cyclone dataset, depicting the mean climatological patterns that will then be decomposed into the clusters in Sect. 3.2. Fig. 1 depicts the density of all Mediterranean cyclones at three stages of their evolution: cyclogenesis, peak intensity, and lysis. Cyclogenesis regions are tightly linked to topography, while cyclones tend to peak over open waters and decay along the eastern coasts and over land (Alpert et al., 1990, Trigo et al., 1999, Campins et al., 2011).

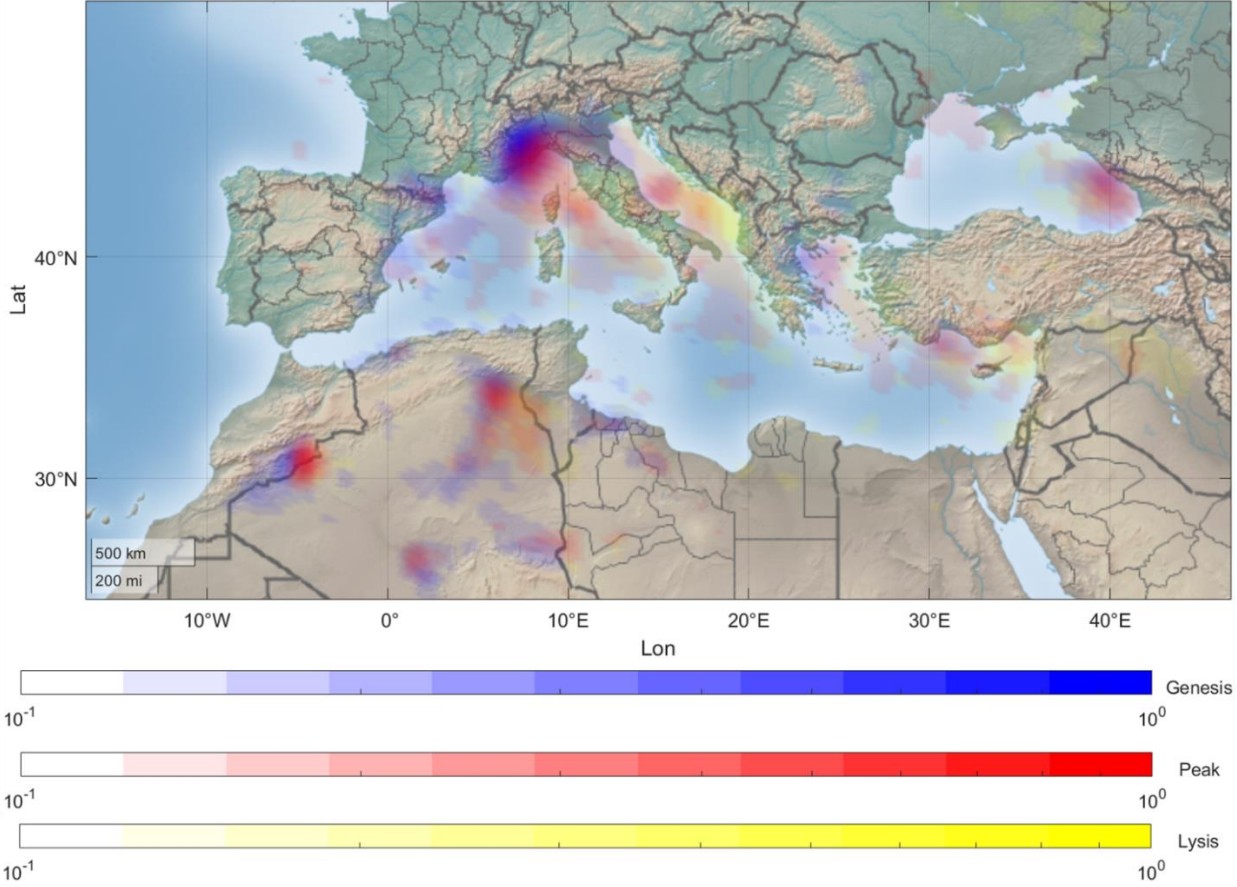

**Figure 1: Geographical climatological density of all 3190 cyclones at their timing of cyclogenesis (blue shading), peak intensity (red shading), and cyclolysis (yellow shading), for 1979-2020. Each cyclone location is gridded with an impact radius of 100 km. The color scale is logarithmic, i.e. dark patches stand for tenfold the density of the lighter patches, and the values are normalized frequency (% of all cyclones in 100 km cells).**

Taking a cyclone-centered perspective, Fig. 2 shows a composite mean of atmospheric fields around all the analyzed MCs, as well as the number of MCs per year and their average propagation velocity. The upper-tropospheric isentropic PV confirms the typical synoptic setting of MCs in the form of a westward-tilted upper-level trough compared to the cyclone center. The warm and cold sectors are evident from the 2-m temperature anomaly field. The asymmetric wind structure is portrayed by the 10-m wind speed composite, exhibiting a maximum to the south of the cyclone center, as well as another maximum further in the northwestern corner, suggesting the imprint of potential parent Atlantic cyclones in some of the cases (Fig. 2c). The composite precipitation displays maxima just to the north of the cyclone's center. Annual cyclone occurrences (Fig. 2e) show a weakly increasing trend, though only statistically significant if evaluated for the smoothed time series. On average, MCs propagate towards the east with a weak poleward component (Fig. 2f), at a 3.5 m/s mean speed. These results agree with multiple past climatological studies addressing MCs (Flaounas et al., 2015, Raveh-Rubin and Wernli, 2015, Flaounas et al., 2021), emphasizing the robustness of the cyclone detection method.

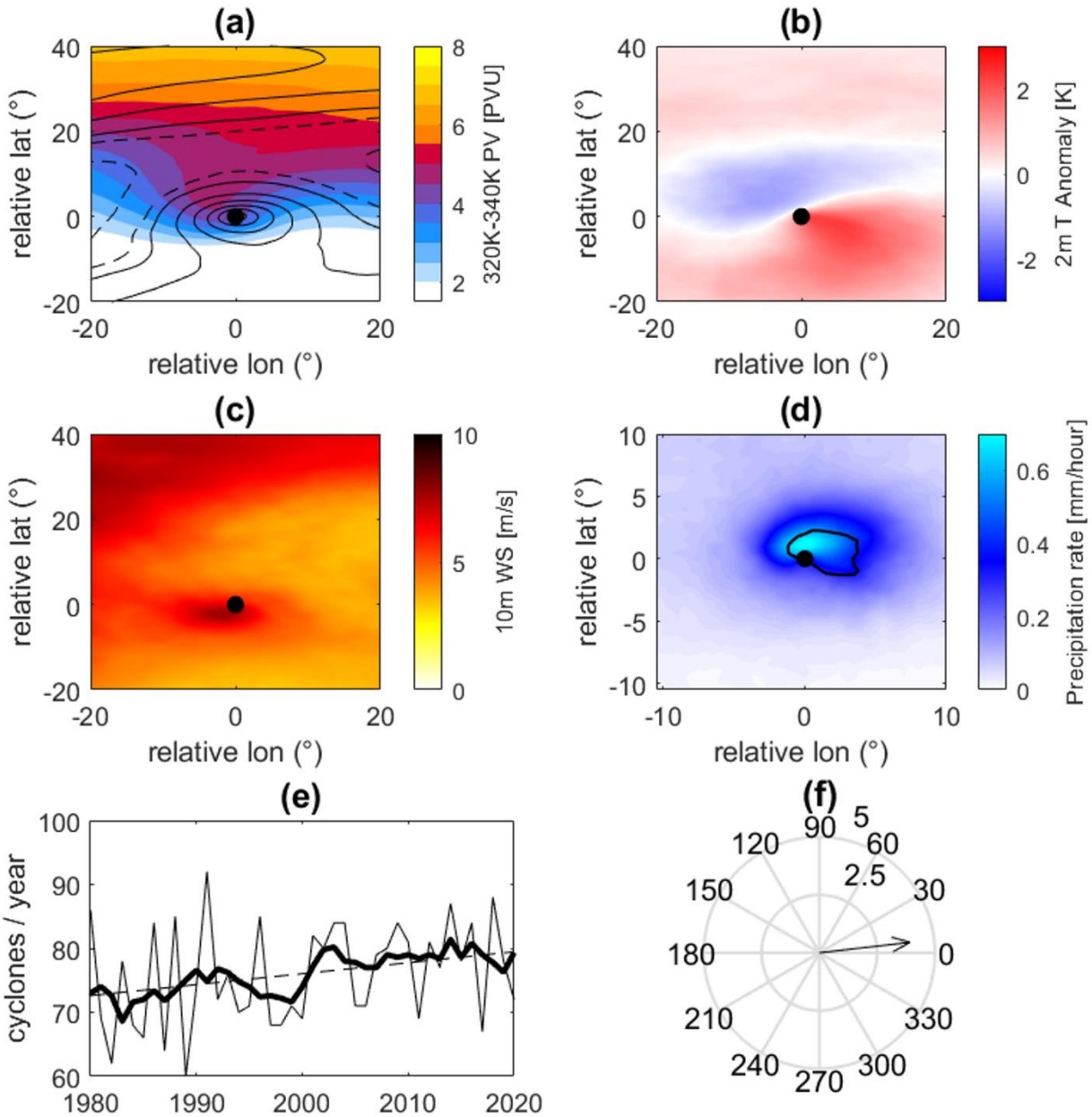

**Figure 2: Composite mean of (a) upper-level (320-340 K averaged) PV (PVU, shading) and SLP (unlabeled contours, at 2-hPa intervals, dashed for values above 1015 hPa), (b) 2-m temperature anomaly from the monthly local climatology (K), (c) 10-meter wind speed (m/s), (d) total (mm/hour, shading), and convective (0.2 mm/hour, black contour) precipitation rates, (e) annual MC occurrences, with a 5-year smoothing (thick line) and linear best fit to the data, and (f) the mean velocity of the cyclone center (m/s). Note the smaller domain in (d). These climatological patterns are used as a reference to establish the statistical significance of the cluster composites and evaluate the variance within the cyclone population.**

## 3.2. SOM classification

The SOM analysis produced 9 remarkably distinct clusters, shown in Fig. 3 with the composite mean PV and SLP for each. The wide variety of PV formations driving MCs spans from cluster 1 having the strongest PV signal to cluster

6 showing the cyclone center furthest from the stratospheric PV reservoir. Various stages of Rossby wave evolution are captured, including cyclonically (8), anti-cyclonically (5) breaking PV streamers and the combination of both (2), as well as cut-offs (3, 9), heat lows (6), and daughter cyclones forming adjacently to synoptic cyclones (7). The frequency of objectively defined PV streamers and cut-offs are composited in Fig. C1, confirming their high relevance also when not directly evident from the composite mean PV. For example, streamers and cut-offs are both most frequent for cyclones of cluster 6 as well as in cluster 9, evidently under ridge conditions with a streamer and/or a cut-off closer to the cyclone center. The relevance of PV streamers is confirmed by their high occurrence under clusters 5 and 8 (both 94%) which also possess a streamer-like composite mean pattern. By construction, the similarity among individual members is larger within clusters than across clusters. Indeed, the cluster means are very distinct from one another, with their position in the SOM map indicating a higher similarity with neighboring clusters compared to distant ones (considering their hexagonal organization, Fig. A1). While clusters 4 and 5 are still very different from one another, the differentiation between clusters 1 and 4 is subtle, but meaningful nonetheless (Figs. 3, C3). The separation of clusters 1 and 4 indicates a different orientation of the dynamical tropopause, hinting towards a different stage of lee-cyclogenesis. This point will be discussed further along with the cyclone features. Together, the SOM map successfully forms a catalog of distinct PV formations driving MCs.

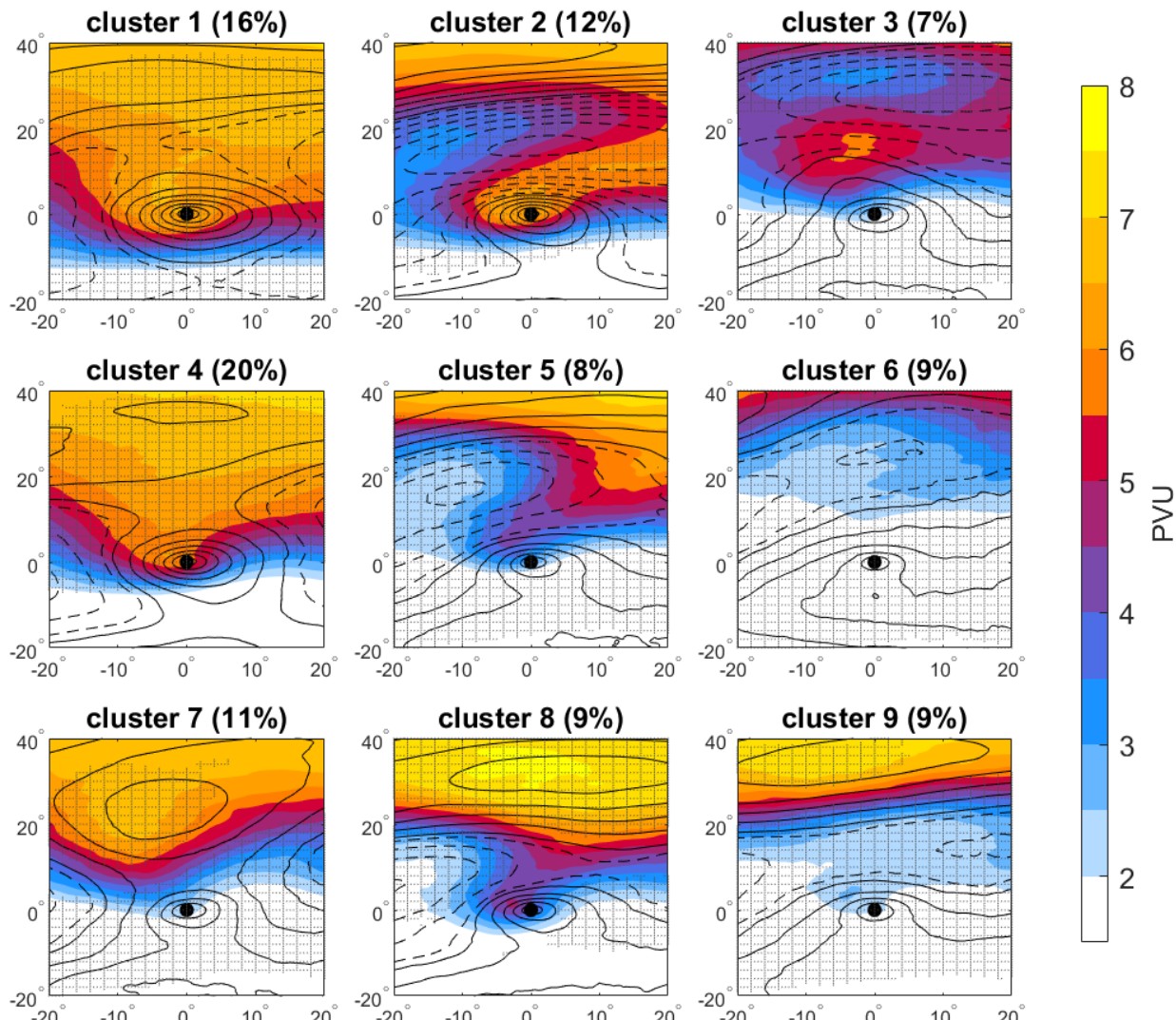

**Figure 3: Cyclone-centered cluster composites of upper-level (320-340 K averaged) PV (PVU, shading) and SLP (black contours at 2-hPa intervals, dashed above 1015 hPa). Stippling indicates a 99% significance level of the PV field concerning the total cyclone average (Fig. 2a). The mean frequency of each cluster out of all cyclones considered is given in the title.**

Addressing the variance in SLP, some cyclones have a smaller impact radius (clusters 5, 6, 8, 9) as expected from their short-wave PV disturbance, while cyclones in clusters 1, 2, and 4 where a long-wave perturbation is portrayed have much greater extent. A surface anticyclone is seen under low PV values (clusters 2, 3, 5, 6, and 8) in different orientations relative to the MC.

The classification is based on upper-level PV to distinguish between the adiabatic, large-scale drivers of cyclone genesis and deepening. The diabatic contribution is often perceived as an independent contributor to cyclone deepening and can be diagnosed by a positive PV anomaly in the mid-troposphere. Fig. 4 shows the composite vertical cross-section of PV in each cluster in the zonal direction, showing that not only the upper-tropospheric PV varies across clusters, but significant differences in the PV distribution down to the surface are also clear. Here, the low/mid-tropospheric PV is generally higher in clusters with more pronounced upper-level PV anomaly, suggesting a relationship between the two PV maxima. Thus, the classification by upper-level PV also sheds light on non-conserved PV in the low- and mid-troposphere, as some clusters (1, 2, 4, 8) show a low-level PV signal, while others do not (3, 5, 6, 7, 9). Interestingly, clusters 3 and 7 show large PV values near the surface, but none in the mid-troposphere. This

feature may be the result of boundary-layer instability or friction over land and/or high orography (Adamson et al., 2006), however, the sources of these PV signals need to be further investigated. A detailed 3D PV analysis is presented in Figs. B1 and B2.

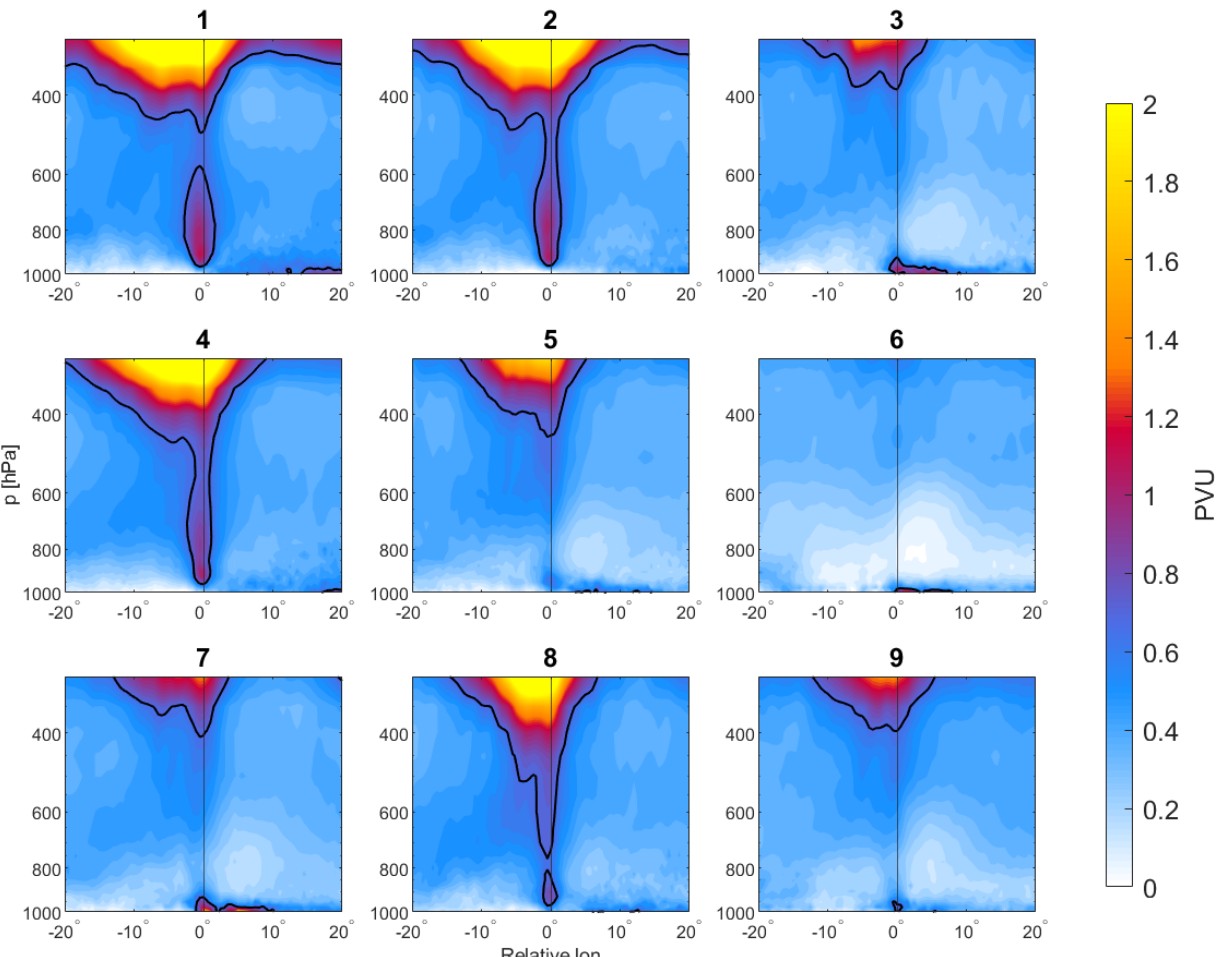


**Figure 4: Zonal vertical cross-section composites through the center of the cyclones, by cluster. PV is shaded (PVU) with 0.7 PVU denoted in black contour. Note the different color scale compared to Fig. 3.**

The anomalies of the composite PV patterns and 10-meter winds are now examined together with 300-hPa wind speed
(Fig. 5). The large magnitude of PV anomalies in clusters 8 and 2 (>2.5 PVU) may be attributed to initially anti-cyclonically breaking PV streamers, later forming cyclonically tilted tips. This formation is accompanied by a double jet structure on the 300-hPa level, prominently for cluster 2. This setup allows the PV streamer to penetrate the furthest equatorward while sustaining large amplitudes above the cyclone, generating the largest upper-level PV anomalies that in turn induce the strongest low-level PV anomalies (Fig. 4 and B2) and surface circulation (Figs 5 and 12). Also
noteworthy are the negative PV anomalies indicative of AWB in clusters 5 and 2, and for CWB in cluster 8. Clusters 3 and 9 show positive PV anomalies above the cyclone center with negative anomalies to the north, indicative of cut-off and/or streamer formations under ridge environments. More details on the PV streamers and cut-offs and their variance across the clusters are included in Appendix C (Fig. C1).

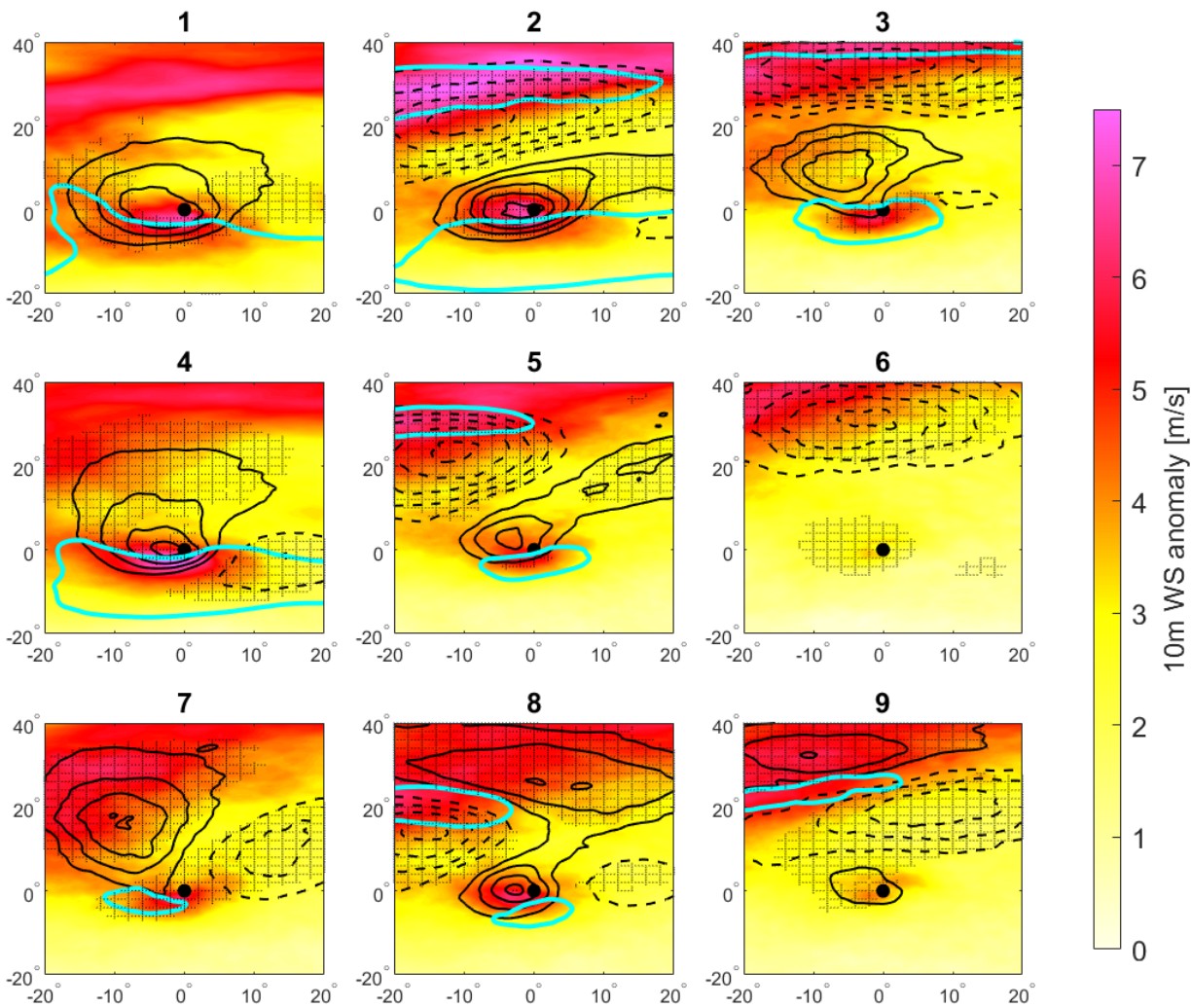


**Figure 5: Horizontal composites of upper-level PV anomalies (0.5-PVU intervals, black contours, dashed for negative) and 10-meter wind speed anomalies (shading) with respect to the local monthly climatology. Overlaid is the corresponding composite of the 30 m/s isotach on the 300-hPa geopotential surface (cyan contour, roughly denoting the subpolar and subtropical jets). Stippling denotes regions where the PV anomaly is statistically significant in the 99% confidence level**

**compared to the total average.**

### 3.3. Cyclone characteristics

The seasonal distribution of cyclones varies sharply by cluster (Fig. 6). Clusters 1, 2, and 4 prominently dominate the winter months, while clusters 6 and 9 mostly occur in summer. The other clusters occur mostly in the transition seasons, some more in spring (3, 7), others peaking in autumn (8), and cluster 5 appears rather equally during both.

Note that despite their high frequency, only 3 winter clusters are obtained, compared to 4 clusters for the transition seasons. This is due to the larger variety of PV features in spring and autumn compared to winter when broad troughs dominate the region. Of course, the seasonal separation among clusters is a result of the choice of the classifying parameter, taking the absolute values of PV, which varies by season due to the choice of the isentropic layer and the equatorward migration and strengthening of the jet in winter. Alternatively classifying by PV anomalies would smear

out some of these differences (not shown). We choose to employ the full PV values and by this also distinguish among the seasons more clearly.

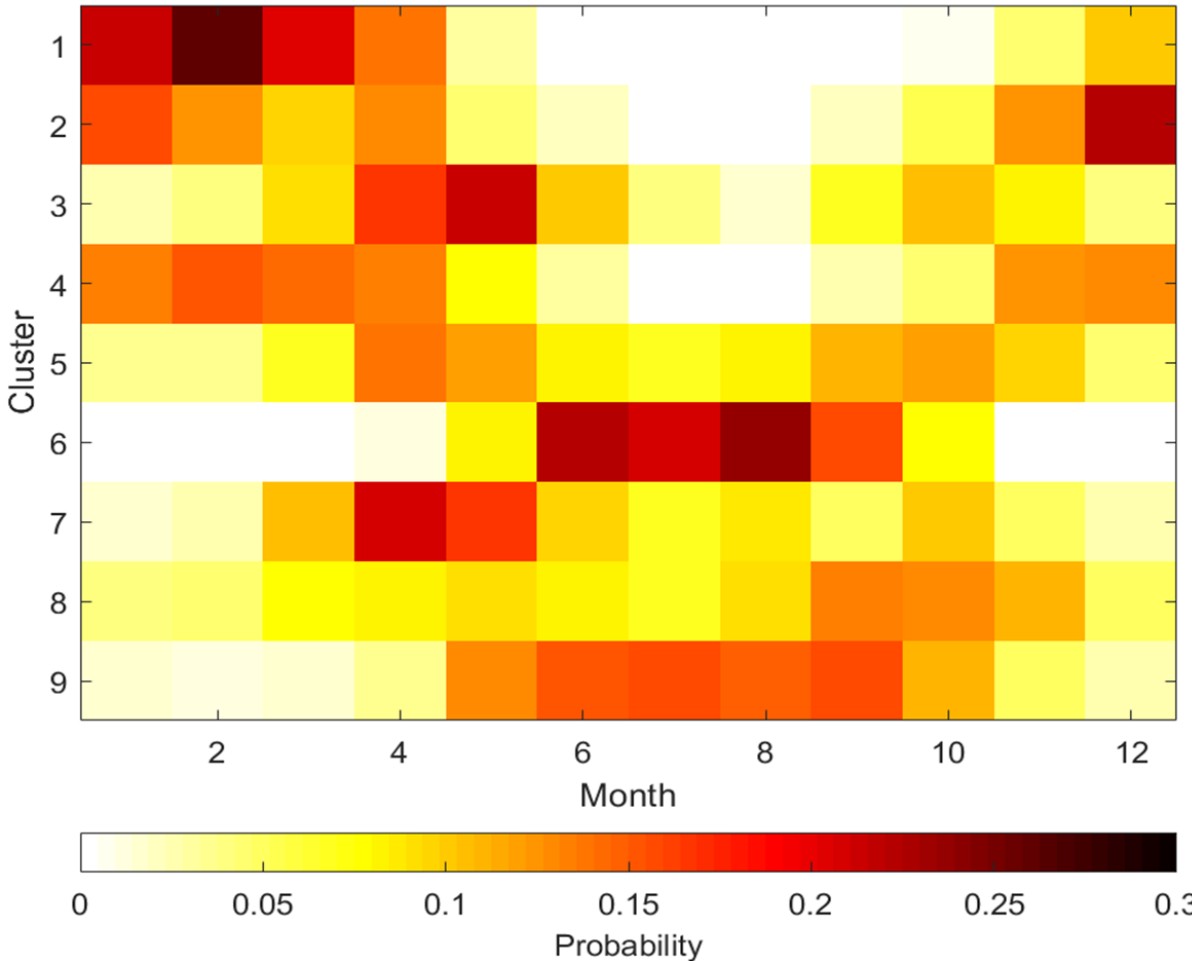

**Figure 6: Monthly distribution of the cyclones in each cluster. The month of each cyclone peak is given on the x-axis, and the corresponding relative frequency is given in shading. The probabilities are normalized relative to the cluster size, i.e., each row sums up to one.**

Examining the geographic distributions by cluster (Fig. 7) reveals highly localized clusters (1, 3, 5, 6, 8) alongside broadly distributed ones (2, 4, 7, 9). Cluster 6 cyclones reside almost exclusively on the African continent, supporting their identification as African heat lows (or Sharav lows), while cluster 1 appears to mainly correspond to Genoa cyclones. Clusters 2 and 4 also originate in the lee of the Alps, however, their peak locations vary greatly, favoring open maritime environments. Clusters 3, 7, and 5 share strong signals near the Atlas, though the latter occasionally form in the Black Sea. Regarding the separation between clusters 1 and 4, the results suggest that cluster-4 cyclones tend to peak further away from topography (as in stage B cyclogenesis) compared to cluster 1 (stage A cyclogenesis). This interpretation will be further examined by their frontal structure in Sect. 3.4. While these geographical-seasonal relationships are well known for MCs (Flaounas et al., 2022), we also detect cyclones appearing in less common locations and seasons, hinting that from an upper-level perspective, dynamically similar cyclones may appear in substantially different spatiotemporal settings. For example, the rare occurrence of a cluster-6 cyclone centered above Italy was co-located with an extreme heat wave that struck Poland and the Balkans on August 2015, leading to the warmest summer recorded in the region since the 1980s (Krzyżewska et al., 2019). Likewise, one of the unusual occurrences of a cluster-2 cyclone near Cyprus resulted in the 2013 Alexa storm, delivering record-breaking snowfall across the Middle East (Sawalha, 2014, Hochman et al., 2022b). Thus, the power of the present analysis is that it is geographically unbound and focused purely on PV dynamics.

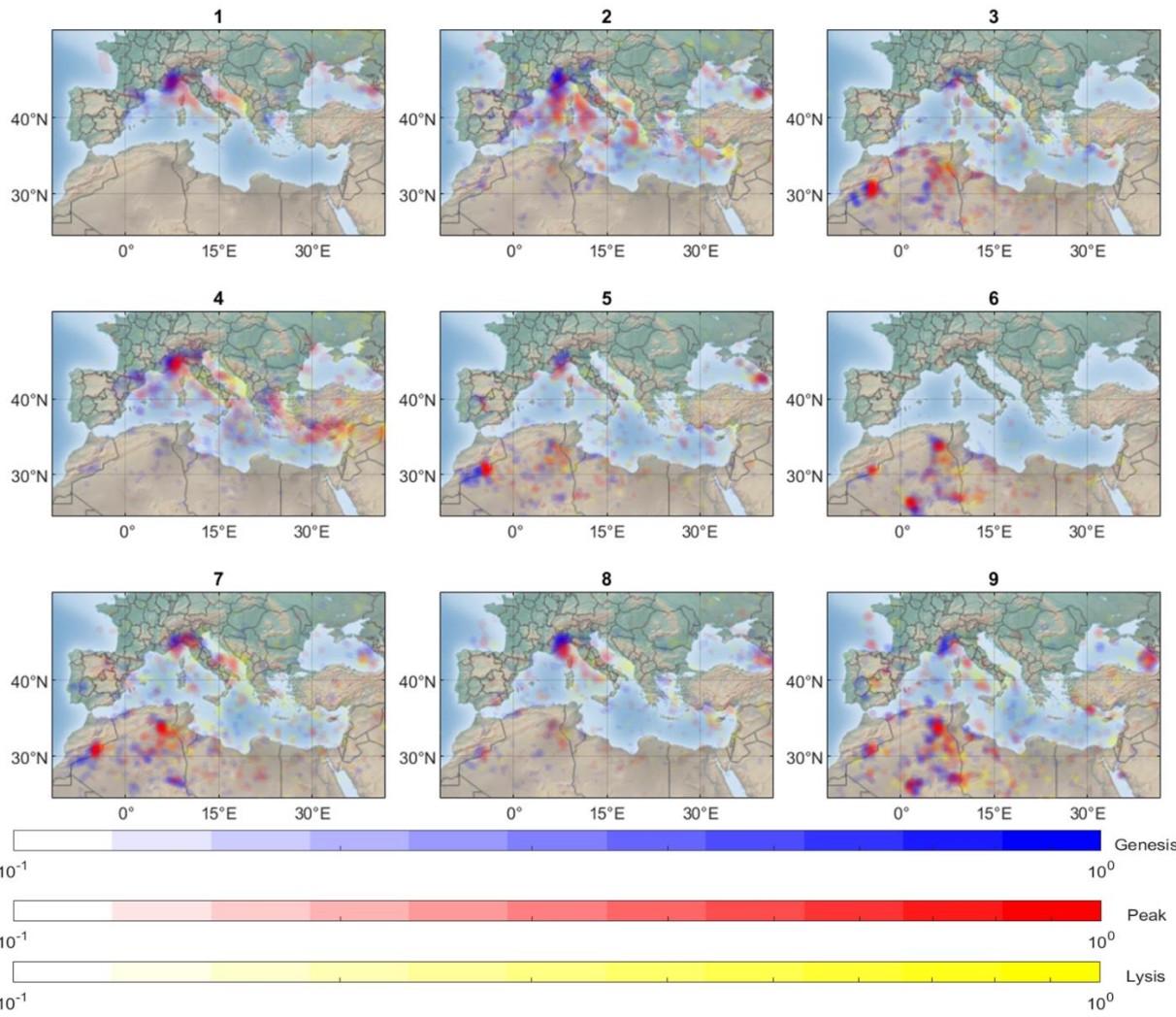

**Figure 7: as in Fig. 1, but for each of the clusters, with frequency normalized separately for each cluster.**


An important aspect of MCs, and particularly challenging to predict, is the tendency of a given cyclone to deepen rapidly and become explosive, i.e., cross the threshold of 1 Bergeron at any point in its lifetime. The distribution of the cyclone maximum 24-h deepening rate by cluster is shown in Fig. 8. Clusters 1, 2, and 4 are the most "explosive", with a fraction of cluster-5 and cluster-8 cyclones also exhibiting explosive systems under Rossby wave-breaking

scenarios. Differently, clusters 6, 3, and 9 tend to deepen more slowly, primarily attributed to the dry environment where these cyclones develop, devoid of humidity fluxes to sustain significant precipitation and latent heat release. Negative Bergeron values indicate tracks that peaked at stages earlier than 12 hours following cyclogenesis, and are most prominent in cluster 1, again suggesting the rapid and early geostrophic adjustment stage A as the cluster's dominant deepening mechanism. This may also be an artifact of Atlantic cyclones entering the domain while becoming

shallower, though only a few such tracks were detected. Cluster 4 also shows a significant number of negative values, in agreement with the notion that most stage B cyclones initiate with a stage A deepening phase yet manage to propagate far enough from the topography and deepen further by releasing baroclinic instability. Thus, while the minimum SLP of cluster 4 cyclones appears to correspond to stage B, the maximum deepening rate, as expected from theory, often occurs during their stage A phase. The same holds for cluster 2 cyclones, to a lesser extent.

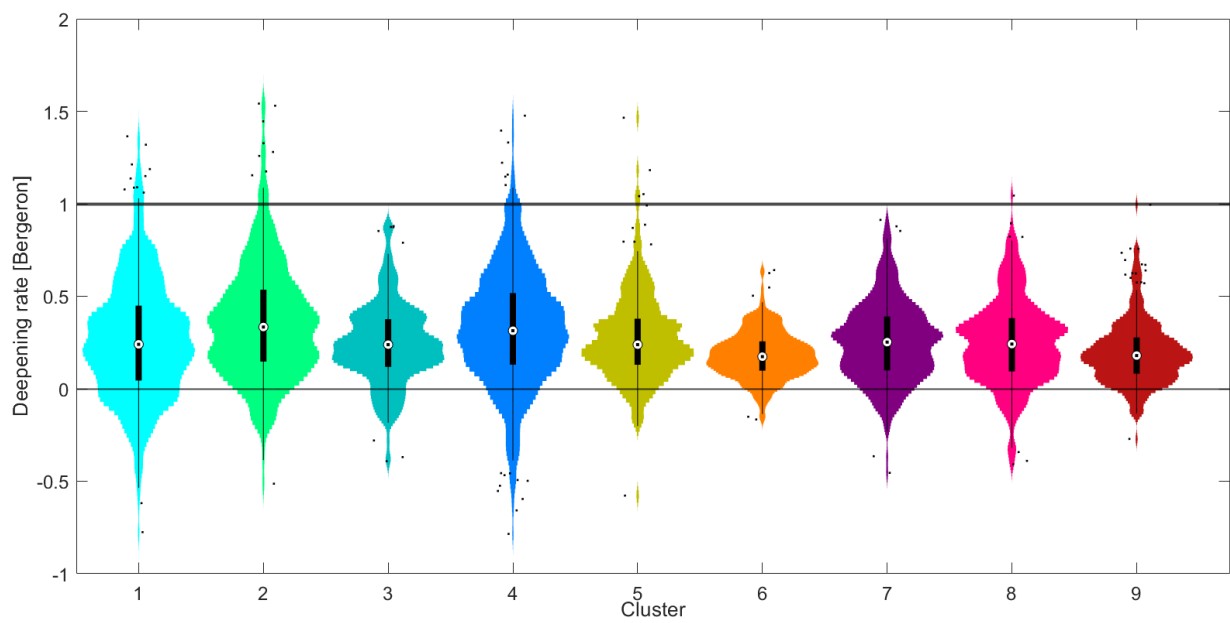


**Figure 8: Box and distribution plots of maximum 24-h deepening rate (Bergeron, 1 denotes the minimum threshold for an explosive cyclone). Encircled dots denote the median values, while the box covers the $\pm 25$ percentiles. Whiskers denote 97.5 percentiles and define the limit for the outliers, denoted by small dots.**

Throughout their lifetime, the cyclones generally propagate to the east with an average speed of 3.5 m/s (Fig. 2, panel f). Fig. 9 shows the cluster-mean anomalies in the cyclone propagation velocity relative to the total average, as well as the radial distribution of the members composing these means (right panels in Fig. 9). When broken down into clusters, coherent positive and negative anomalies of the order 1-2 m/s are evident. These anomalies are relatively consistent throughout each track and among each cluster, as seen by the radial histograms. Some clusters (1, 3, 4, 7)

propagate eastward faster than average, while others (2, 5, 6, 8, 9) appear more static. Stationary cyclones pose a significant threat to extreme weather occurrences, by influencing local regions for longer periods. This aspect will be discussed further when examining surface impacts in Sect. 3.4.


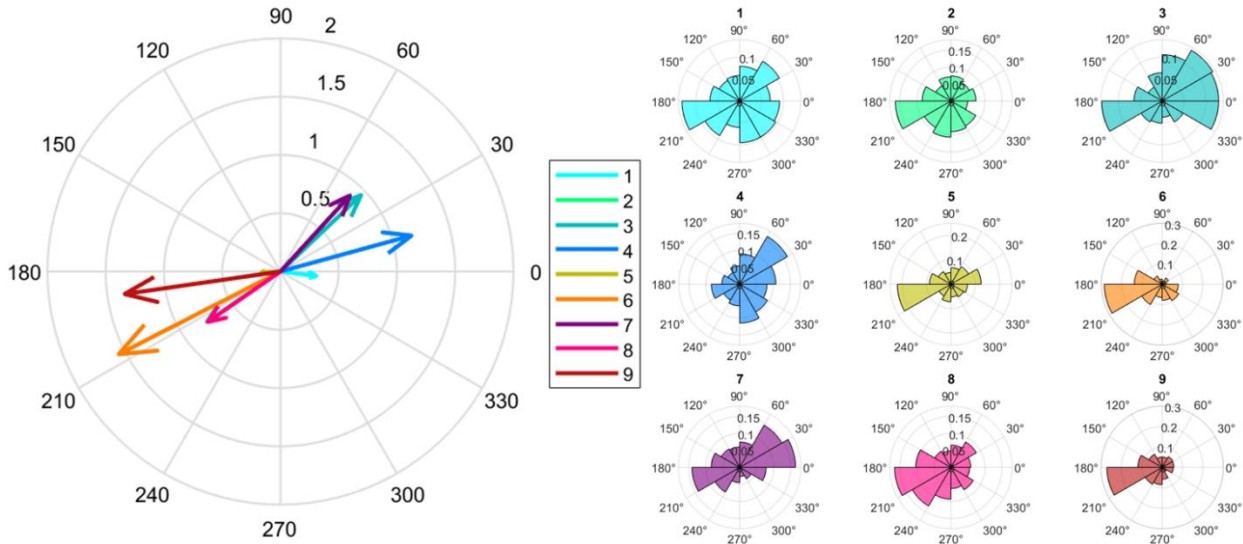

**Figure 9: Distributions of cyclone propagation anomalies averaged throughout the cyclone track, by cluster. Left: polar vectors indicating the cluster mean deviation from the total average shown in Fig. 2f (colored vectors). The azimuth and radial labels show the angle (0° marks eastward propagation) and magnitude (m/s) of the anomalies. Note that clusters 2 and 8 share a similar vector space, making cluster 2 vector hardly visible. Right: polar density plot illustrating the anomaly distributions of the angle composing each colored vector to the left.**

### 3.4.    Surface Impact

The emerging differences among clusters in terms of their PV distributions, as well as their variability in terms of spatiotemporal occurrence and cyclone characteristics, indicate that different clusters are linked to different kinds of surface impacts. Here we examine composites of meteorological hazards (2-m temperature, 10-m winds, and precipitation) by cluster to examine the range of possible surface impacts and their mean cyclone-relative distribution and area of impact for the timing of peak cyclone intensity. Indeed, the cluster composites of 2-m temperature anomalies depicted in Fig. 10 are strikingly different. Cold cyclones (clusters 1 and 2) show negative anomalies across the domain, while the opposite is true for North African heat lows (cluster 6). Extended warm anomalies also govern the summer cut-off low (cluster 9). Dominant warm surface anomalies prevail in clusters 3, 5, and 7, providing the cyclonic (positive) surface PV in all these clusters (compare Figs. 4 and 9). Cluster 4, as opposed to cluster 1, shows the development of a warm sector alongside the cold one, again emphasizing the thematic separation between stages A and B of lee-cyclogenesis. The daughter cyclones attributed to cluster 7 appear to form along the frontal system of the parent cyclone, evidently preferentially in the warm sector. Given the large variation from the averaged MCs presented in Fig. 2, classification by upper-level PV also enables a distinction among meaningful surface temperature distributions, with clusters 3, 4, 5, 7, and 8 showing a frontal structure with both warm and cold anomalies, clusters 1 and 2 having a pronounced cold sector, and clusters 6 and 9 marked by a warm anomaly throughout, without an evident frontal structure. The frequency of warm and cold fronts detected around the cyclone is derived using composites of the identified front objects (Fig. C4). Consistently, fronts are more likely to be identified in clusters 1, 2, and 4 than in clusters 6 and 9, and their spatial alignment varies strongly.

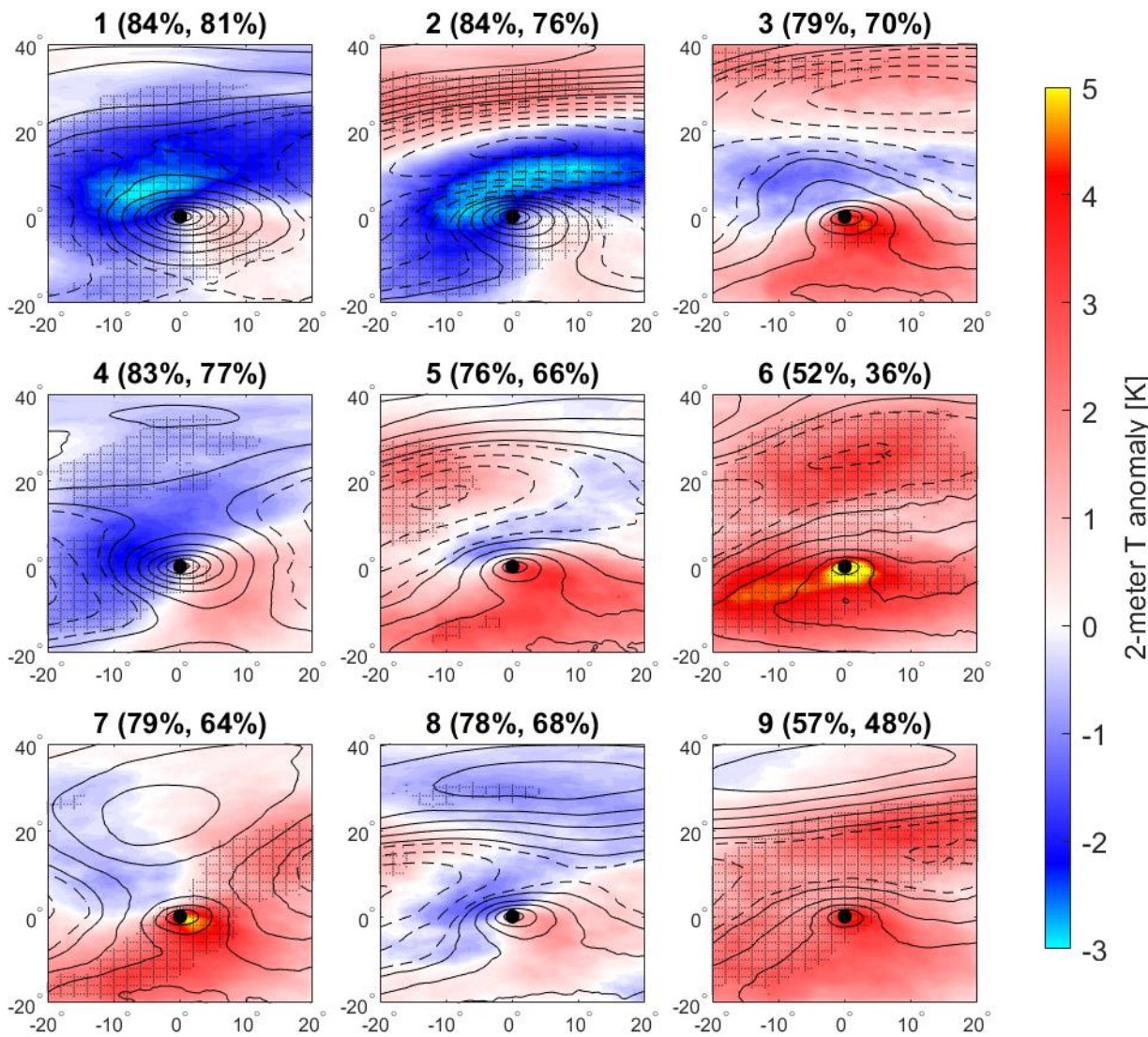


**Figure 10: Composites of 2-m temperature anomaly from the local monthly climatology (K, shading) by cluster. Stippling denotes the 99% confidence level compared to the averaged field shown in Fig. 2b. Cold and warm front occurrence frequencies in $\pm 5^\circ$ degree box around the cyclone are given in the title. Reference composite SLP contours are shown as in Fig. 3.**


Precipitation variability is a key feature when classifying MCs, and in Fig. 11 the precipitation by clusters is illustrated, in terms of convective and large-scale contributions. Generally, winter cyclones (e.g., clusters 1,2, and 4) precipitate the most, primarily due to large-scale uplifting. The Autumn cluster 8 also exhibits comparable precipitation averages, although with a relatively smaller large-scale contribution. Summer clusters 6 and 9 are drier, especially African heat

lows. The latter, cluster 6, has rain rates smaller than 0.2 mm/hour (on average), and in 92% of the cases has no significant precipitation at all (below 0.1 mm/hour at a range of $\pm 5^\circ$ from the center). Cluster 6 is, however, tightly related to significant convection precipitation anomalies further to the south, attributed to the African monsoon (not shown). Precipitation from cluster-2 and cluster-8 cyclones may be particularly impactful, as these cyclones tend to be more static (compared to the other clusters, see Fig. 9).

The changes in convective precipitation mostly complement those driven by the large-scale dynamics, except for clusters 1 and 8. In cluster 1, a competing effect is evident, as convective precipitation is decreased to the north of the

cyclone where the large-scale contribution is increased, compared to the overall cyclone's mean. This may be due to the downward movement into the lower levels found on the lee-side of the mountains influencing cluster 1 cyclones (i.e., the Alps, primarily). This downward motion is expected to increase stability and hence reduce convective precipitation. However, the large-scale contribution dominates and leads to overall increased precipitation, likely by WCBs (Fig. C2). In cluster 8 the opposing competitive effect is evident. Increased convection overlaps a large-scale decrease in precipitation. This feature might also be linked to the topography, which enables the southward extension of the PV streamer, generating convection near the tip. The decrease in the large-scale precipitation corresponds to the anticyclonic wave-breaking fraction of the streamer, while the cyclonic tip triggers convection on lower levels. Overall, cluster 8 shows the strongest precipitation rates of all non-winter modes, again highlighting a unique autumn contribution to precipitation, and suggesting the importance of the double wave-breaking pattern (Anti-cyclonic followed by cyclonic wave breaking) to the intensity of MCs (see Fig. B1).

The presence of warm conveyor belts and their spatial orientation is highly influential on precipitation patterns. WCB mask composites for the mid-troposphere are presented in Fig. C2, with large variations between the clusters. Interestingly, while WCBs indeed dominate clusters with clear wet deviations (both large-scale and convective), such as clusters 1 and 4 (but somewhat less in cluster 2), WCBs also dominate clusters 3 and 7 while their precipitation amounts are low, represented by negative deviations. The latter is likely due to the large fraction of cluster-3 and cluster-7 cyclones over land in northwest Africa, where moisture sources for precipitation are limited. In these cases, dust is likely transported along dry WCBs (Fluck and Raveh-Rubin, 2023, Rousseau-Rizzi et al., 2023). Notably, a cluster 7 cyclone peaking on April 15th, 2005 traveled from North Africa towards Greece, generating an extreme dust storm (Kaskaoutis et al., 2008).

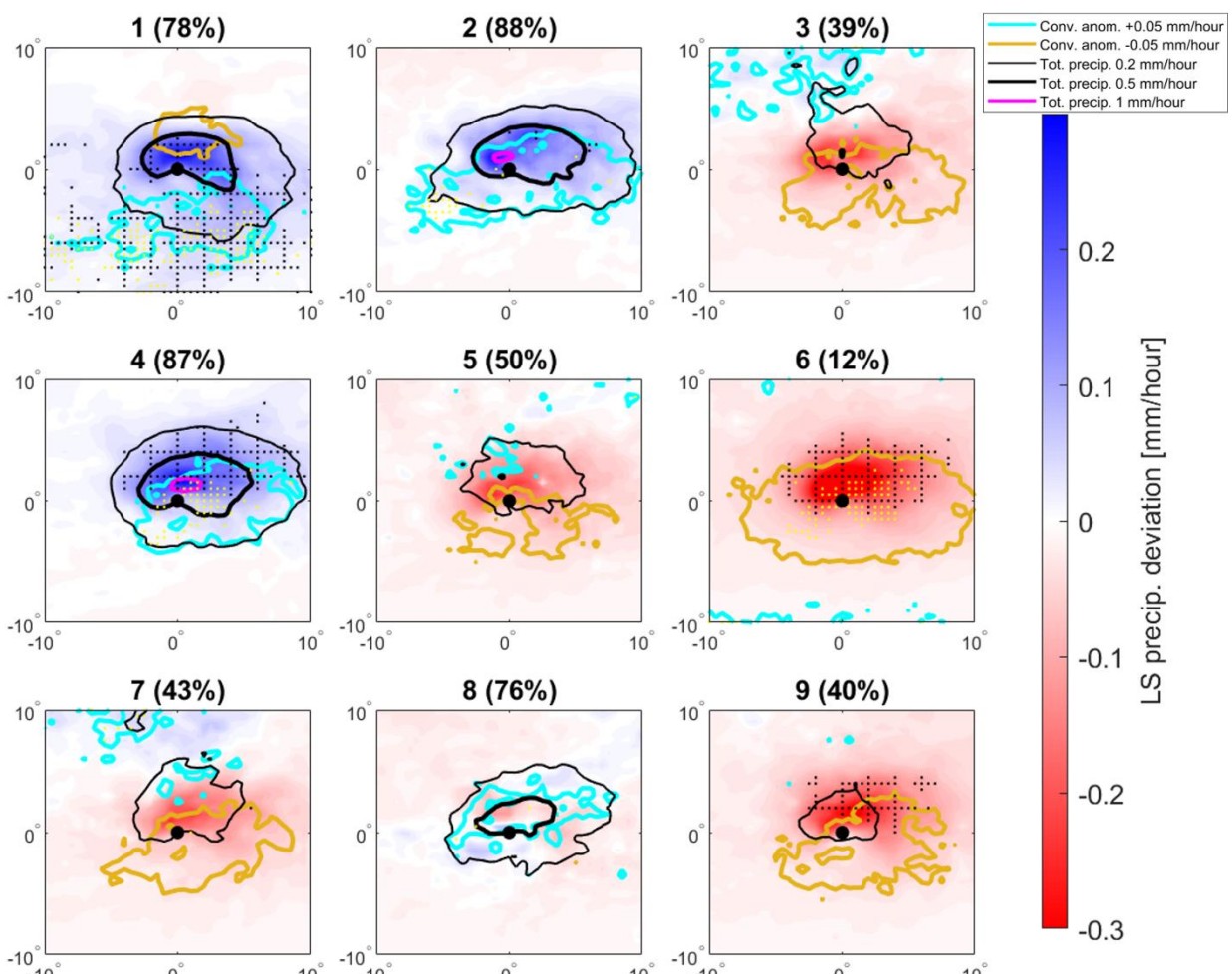

**Figure 11: Precipitation deviation with respect to the cyclone mean in Fig. 2d, shown separately for the large-scale (mm/hour, shading) and convective (±0.05 mm/hour, cyan and yellow contours for positive and negative) precipitation deviations. The black contours show the absolute, cluster-averaged daily precipitation rate (sum of large-scale and convective precipitation, 0.2 and 0.5 mm/hour), while rain rates exceeding 1 mm/hour are shown in magenta. Stippling denotes regions where the large-scale response is statistically significant in the 90% confidence level compared to the total mean and yellow dots denote the corresponding significance level for the convective response. Note the domain size is smaller than the other composites. In brackets is the cluster frequency of cyclones with a total precipitating rate greater than 0.1 mm/hour anywhere within ±5 degrees (i.e., 10°X10° box around the cyclone).**

The response of the 10-meter wind speed is presented in Fig. 12. While summer clusters (6,9) show significantly weaker winds revolving the cyclone compared to winter cyclones (1,2,4), the response of the other clusters is mostly statistically insignificant. Interestingly, AWB+CWB cyclones (cluster 2) demonstrate the most intense cyclonic motion at the surface, despite being shallower than the lee-cyclones 1 and 4. This apparent contradiction is resolved by linking the enhanced upper-level PV anomaly in cluster 2 to surface winds, as further discussed in Appendix B, highlighting the importance of the topographically induced RWB. The PV streamer acquires an anti-cyclonic tilt over the wind side of the Alps, and cyclonic tilting/ wrapping on the lee-side, which extends far to lower levels, and intersects with low level PV to form a PV tower.

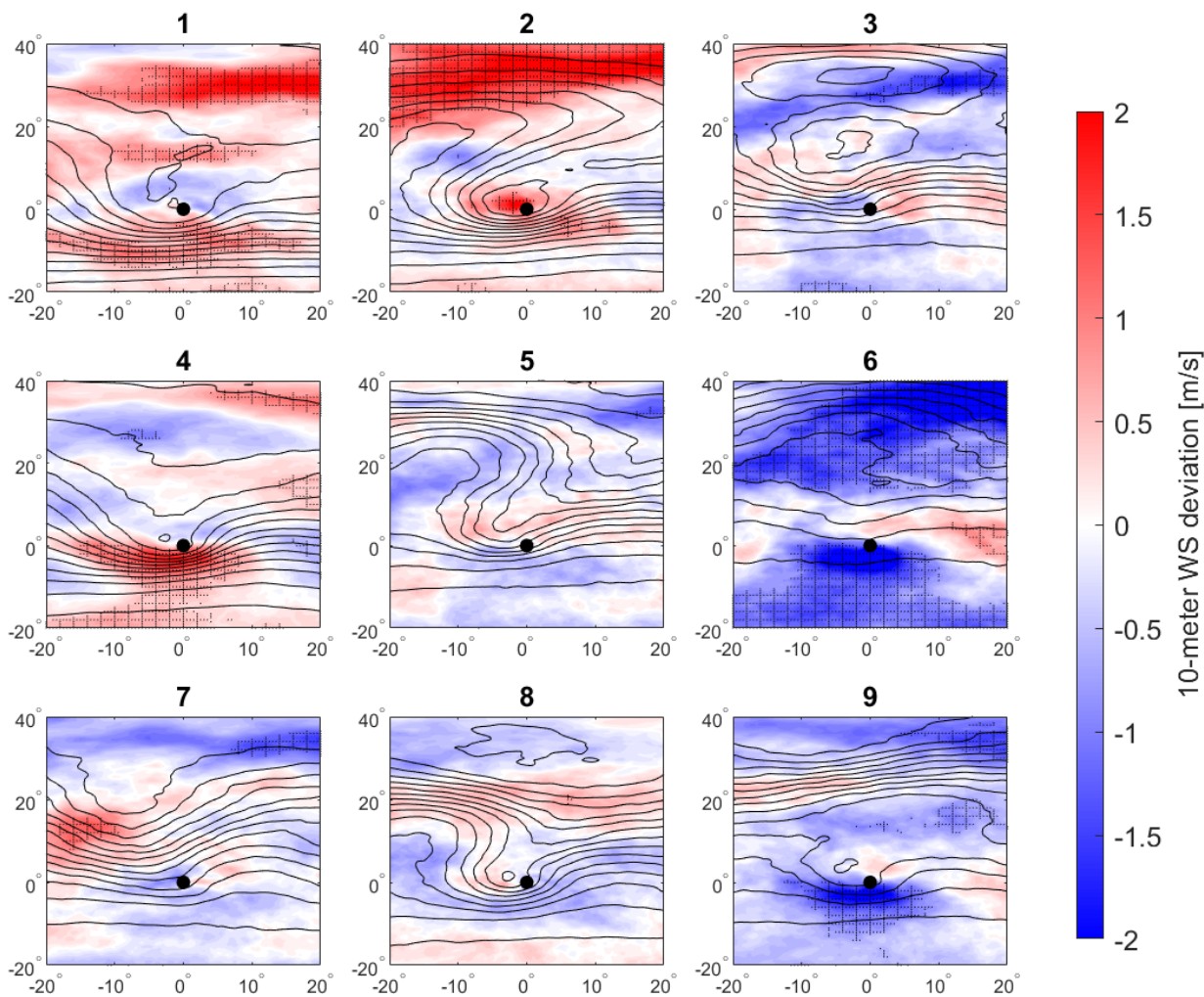

**Figure 12: Composites of the deviation of 10-meter wind speeds with respect to the total cyclone average. Reference composite PV contours are shown as in Fig. 3 (black contours). Stippling denotes regions where the wind speed response is statistically significant in the 99% confidence level compared to the total cyclone mean.**

### 3.5. Trends and predictability

To explore the potential of the PV clusters framework to improve the understanding of MC predictability, we examine the cluster composites of PV at cyclogenesis, namely the initiation time (and location) of the same tracks that build up the cluster. Noting that it takes ~40 hours on average from cyclogenesis to reach the cyclone minimum SLP (not shown), the cluster composites at time 0 resemble the peak-time composites (compare Fig. 13 to Fig. 3). Thus, each track can be readily assigned to its cluster already at the cyclogenesis stage, suggesting high predictability of the cluster type due to the little change of the PV signature between these time steps.

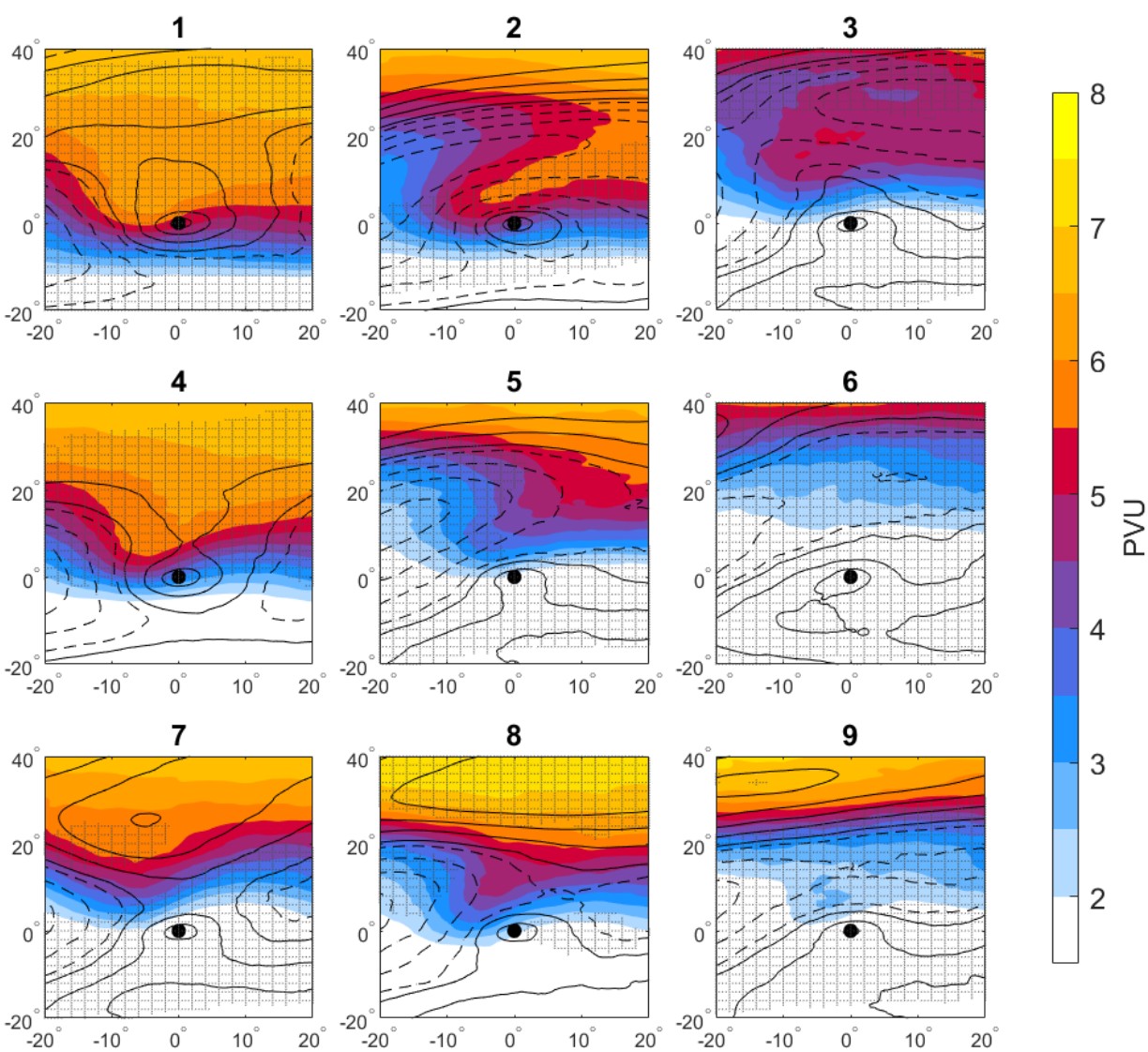

**Figure 13: As Fig. 3 but at the cyclogenesis stage.**

Considering the fundamental differences among cyclone classes, it is useful to evaluate observed long-term trends in their frequency of occurrence. Despite large year-to-year variability, using a 5-year moving mean, significant trends emerge for some clusters. Generally, when considering all cyclones together, a weak increasing trend emerges with 1-2 more cyclones per decade (Fig. 2e). The decomposition into clusters, however, reveals some competing trends which make up this weak signal (Fig. 14). A significant decrease in the occurrence of winter clusters 1 and 4 is evident. At the same time, there is an increase in the occurrence frequencies of summer clusters 6 and 9 and transition-seasons clusters 3 and 5, though the latter two only show significance when computed for the moving mean. These trends agree with the expected poleward shift of the storm track (Caian et al., 2021), reducing the impact of Atlantic cyclones on the northern Mediterranean winter and expanding tropical influences poleward in summer (Cook and Vizy, 2019, Tuel and Eltahir, 2020). These changes lead to the recent decrease in precipitation and increasing heatwaves, reported throughout Western Europe and the Middle East (Lionello et al., 2014). The significance of these trends in reanalysis serves as motivation to investigate future projections of the cluster's frequency by decomposing Mediterranean cyclones into their different types which likely have opposing trends. Furthermore, a significant increase is noted in anomalous occurrences of cluster 6 cyclones (not shown) in spring over Europe, indicating a mechanism of European drying that poses concerning prospects to regional climate.

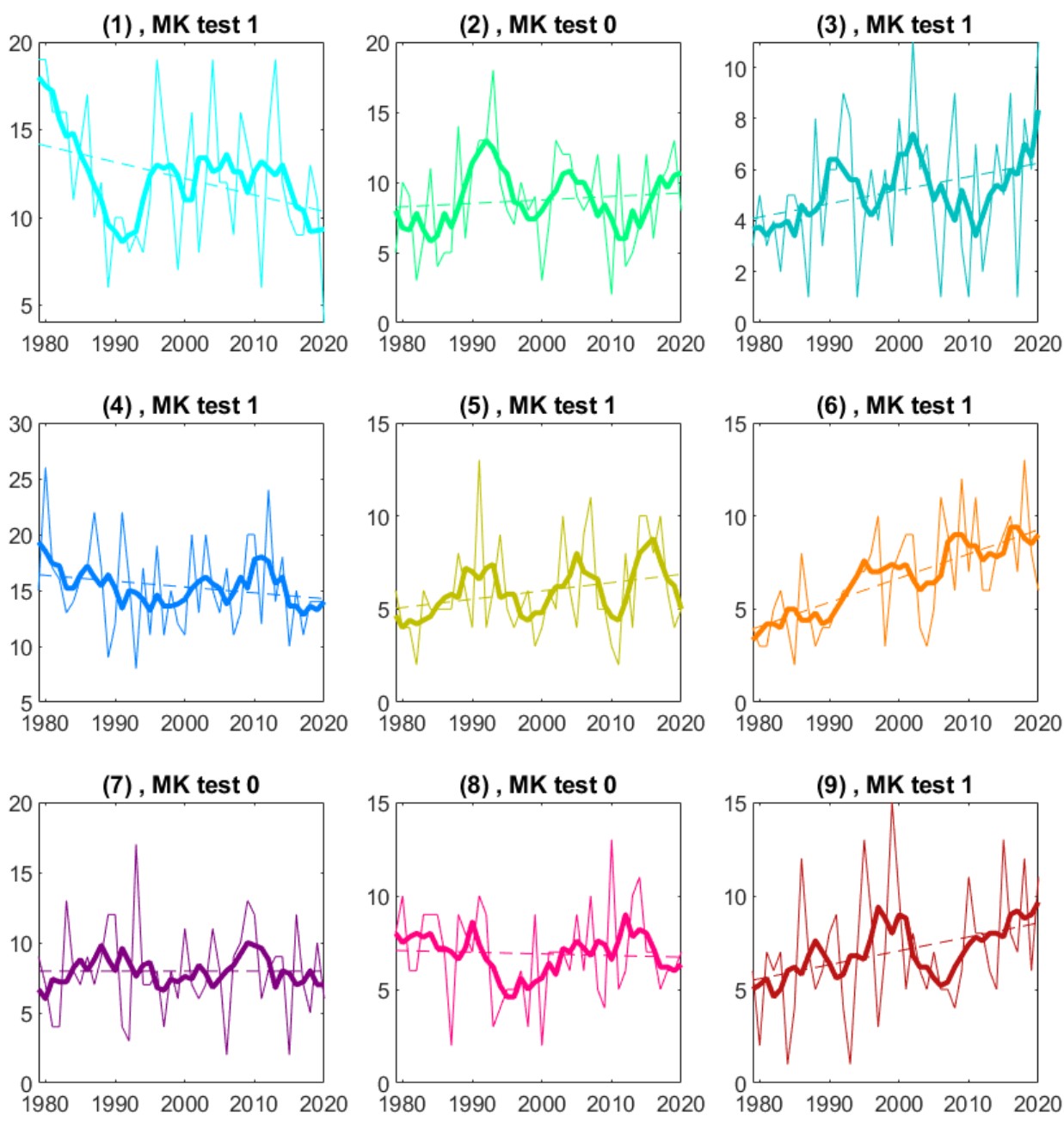

**Figure 14: Inter-annual cyclone frequency by cluster (thin line) and 5-year moving mean (thick line). The long-term trend statistical significance is evaluated with the Mann-Kendall (MK) test for the smoothed data on a 90% confidence level. The MK results (binary, 1 indicates a significant trend) are given in the title. The dashed line shows the linear best fit.**

## 4.    Summary and Discussion

The climatological SOM classification of MCs by their upper-level PV distributions resulted in 9 fundamentally different MC types. Despite classifying a 2D map at the peak time of each cyclone track, each cluster displays significant variability across the 3D space and throughout the cyclone's lifetime. Considered as a whole, distinct surface impacts and cluster features highlight the dominant process driving MCs in each cluster: from daughter and

topographically induced cyclones to heat (thermal) and cut-off lows. A separation between stages A and B of lee-
cyclogenesis (Buzzi and Tibaldi, 1978, McGinley, 1982) is reflected in this analysis, and AWB and CWB
(corresponding to lifecycles 1 and 2, respectively, Thorncroft et al., 1993) are captured as well. While some clusters
are highly influenced by topography, the cyclone-centered perspective enables us to tell apart differently driven MCs
even when sharing similar regional and seasonal amplitudes, and vice versa, i.e., relate dynamically similar cyclones
occurring in different regions or seasons. For example, a "Cyprus low" may appear in similar seasons as cluster 2 or
4, with substantially different weather implications, while similar cyclones in those clusters frequently occur in other
regions but Cyprus. Table 1 summarizes qualitatively the main MC driving mechanism by cluster, based on the
collective information of its geographic, seasonal, thermal, and dynamical features as discussed throughout this work.

| Cluster | (1) Stage A lee-low | (2) AWB+CWB low | (3) Long-wave cut-off low | (4) Stage B lee-low | (5) AWB low | (6) Heat low (Sharav) | (7) Daughter low | (8) CWB low | (9) Short-wave cut-off low |
|---|---|---|---|---|---|---|---|---|---|
| Season | winter | winter | spring | winter | spring/autumn | summer | spring | autumn | summer |
| Genesis | Alps | Alps | Atlas | Alps | Atlas | Atlas | diverse | Alps | diverse |
| Peak | Ligurian Sea | diverse | Atlas | diverse | Atlas | Sahara | diverse | Ligurian sea | diverse |
| Cold sector | extreme | extreme | weak | strong | mild | faint | weak | mild | faint |
| Warm sector | faint | Weak | strong | mild | strong | extreme | extreme | mild | strong |
| Precipitation | heavy, LS+ C- | heavy, LS+ C+ | mild, LS- C- | heavy, LS+ C+ | mild, LS- C- | weak, LS- C- | mild, LS- C- | heavy, LS- C+ | mild, LS- C- |
| Mobility | fast | slow | fast | fast | average | slow | fast | slow | slow |
| Trend | decrease | none | increase | decrease | increase | increase | none | none | increase |

**Table 1: Summary of the main cyclone cluster characteristics. Precipitation is summarized as the mean absolute pattern, large-scale (LS), and convective (C) anomaly signs.**

The SOM illustrates the spectrum of MC variability on a scale ranging from midlatitude winter cyclones (clusters 1
and 4) to subtropical summer heat-lows (clusters 6 and 9), highlighting the different Rossby wave lifecycles in
between. Cluster 5 indicates the dominance of AWB while cluster 8 shows a modified pattern of CWB. A cut-off low
dominates cluster 3, while cluster 7 appears to correspond to daughter lows forming along the frontal system of a
synoptic-scale cyclone (possibly a Mediterranean cyclone of cluster 1 or 4, or an Atlantic cyclone). Cluster 2 includes
some of the most intense MCs, and yields the largest PV anomalies despite being dominated by AWB, usually
associated with the cyclone decaying phase. This highlights the coupling between upper- and lower-PV anomalies
that may be an artifact of PV interaction with orography. Indeed, for a cyclone initiated in the Gulf of Genoa (as
clusters 2 and 8 often do), AWB is expected on the windward side of the Alps, while cyclonic shear is forced on the
lee-side, joined by the Mistral further south, possibly leading to a secondary CWB event. Such an event is presented
in Givon et al. (2021), in which the PV streamer rotates anti-cyclonically on the windward side of the Alps and flows
over the Rhone Valley. It then tilts cyclonically and breaks on the lee-side, forming a spiral that converges toward the
surface cyclone and eventually forms a cut-off low (see Appendix B). The process is accompanied by rapid deepening
of the surface cyclone that peaks during the CWB stage. Similar formations with different magnitudes/wavelengths
appear to form clusters 8 and 2, the most intense clusters in their respective seasons. This double wave breaking pattern
is further discussed in the Appendix. Compared to cyclone-centered classifications performed in other regions on the
globe (e.g., Catto 2018), MCs appear smaller, with larger influences of heat lows and AWB. However, since this is
the first attempt to classify cyclones by their upper-level PV structures, it is yet unknown how the PV patterns are
expected to vary in different geographic regions. Overall, the dynamic classification represents substantially different
MCs that vary in their driving mechanism, with significant implications for weather and climate forecasting.

## 5.    Concluding remarks

Mediterranean cyclones are a major cause of severe weather affecting the lives of millions throughout Europe, North
Africa, and the Middle East. Current understanding of MC predictability is lacking, with an emphasis on predicting
the cyclone location, deepening rate, and distribution and intensity of its meteorological impact (precipitation,
temperature, surface winds). Using a SOM approach, upper-level PV serves to classify cyclones and systematically
evaluate the large-scale patterns driving different cyclone types, with distinct seasonal and geographical distribution,
a characteristic cyclone lifecycle, dynamic and thermodynamic response, and associated hazards. Specifically,

differences in the explosiveness and propagation of each cluster are evident, and a dynamic relation between upper-level PV anomalies and low to mid-tropospheric PV anomalies is suggested.

A caveat in this approach is the indirect link to the contribution of orography to cyclone development. For example, clusters 2 and 8 are influenced by orography. In what may be a unique PV structure for the Mediterranean basin, these cyclones appear to portray an anticyclonically breaking PV streamer that tilts and breaks cyclonically near its tip (Fig. B1 in Appendix B). It is shown that such PV streamers penetrate lower latitudes, creating the largest PV anomalies in both upper- and lower levels, resulting in the most vigorous cyclones with the strongest surface winds and precipitation rates for their respective season. Furthermore, due to their interaction with orography, these cyclones propagate slower than average, making their surface impact even more destructive.

We trace each cyclone track back to initiation time to show that the PV clusters may already be identified at cyclogenesis. This may allow forecasters to identify the cyclone type only by its cyclone-relative PV distribution, with profound implications for the cyclone's expected intensity, thermal structure, precipitation rate, propagation velocity, etc., as shown here throughout. These insights may be useful to constrain the large ensemble spread that often arises in MC forecasts (Binder et al., 2021, Portmann et al., 2020).

We conclude with long-term trends suggesting a future increase in summer heat lows and a corresponding decrease in cold cyclones near the Gulf of Genoa. While the latter has been shown in past studies, in both historical analysis and future projections (Trigo et al., 2000, Zappa et al., 2015, Reale et al., 2022), increasing trends of Mediterranean thermal lows were only reported by future projections (Lionello and Giorgi, 2007), and were not detected in historical data until now. The distinct, and at times opposing trends of different MC types have important implications for understanding and mitigating the future impacts of MCs across the Mediterranean region, and far into Europe, North Africa, and the Middle East. We emphasize the importance of considering the cyclone class when associating current and future impacts to MCs, extending to aspects not considered here, such as hail, dust transport, fires, or compound hazards (Khodayar et al., under review).

The upper-level PV classification yielded a thematic separation between different cyclone-driving mechanisms, unbound by geographical considerations. This method transcends the common geographical-seasonal classifications of MCs, enabling the identification of highly similar events occurring in different seasons and/or geographical regions. We hope this perspective can be used to enhance the understanding of MC's predictability on both weather and climate scales and serve as an insightful approach to classifying extratropical cyclones at large.

**Acknowledgments**

This article is based upon work from COST Action CA19109 "European network for Mediterranean cyclones in weather and climate" (Hatzaki et al. 2023), supported by COST (European Cooperation in Science and Technology, www.cost.eu). We are grateful for the COST Action initiative for the definition of a Medicane, led by Marcello Miglietta, for the insightful discussions, and for providing the list of Medicanes. We further acknowledge Giulia Panegrossi, Valentina Di Francesca and Leo Pio D'Adderio for generously associating the Medicanes to the clusters. This research was partially supported by the de Botton Center for Marine Sciences, by the Israeli Council for Higher Education (CHE) via the Weizmann Data Science Research Center, and by the Weizmann Institute Sustainability and Energy Research Initiative (SAERI).

**Appendix A: SOM analysis**

The SOM is composed of a hexagonal grid topology, with an initial coverage period of 400 steps (~12% of all cyclones, in this case). The initial neighborhood size is set to 2, and the SOM analysis includes 800 iterations. The PV input is rescaled by its total average and standard deviation to range between 0-1 (as in Givon et al., 2021), which helps reduce SOM errors, defined as the total mean squared error (MSE) between the cluster's members and composites, averaged across all clusters. SOM shows weak sensitivity to these initial parameters within a reasonable

range of values (not shown). The number of clusters (N=9) is selected using the elbow method (Liu and Deng, 2020), as the number beyond which a sharp decrease in the SOM learning rate (i.e., SOM error reduction per added cluster) is evident.

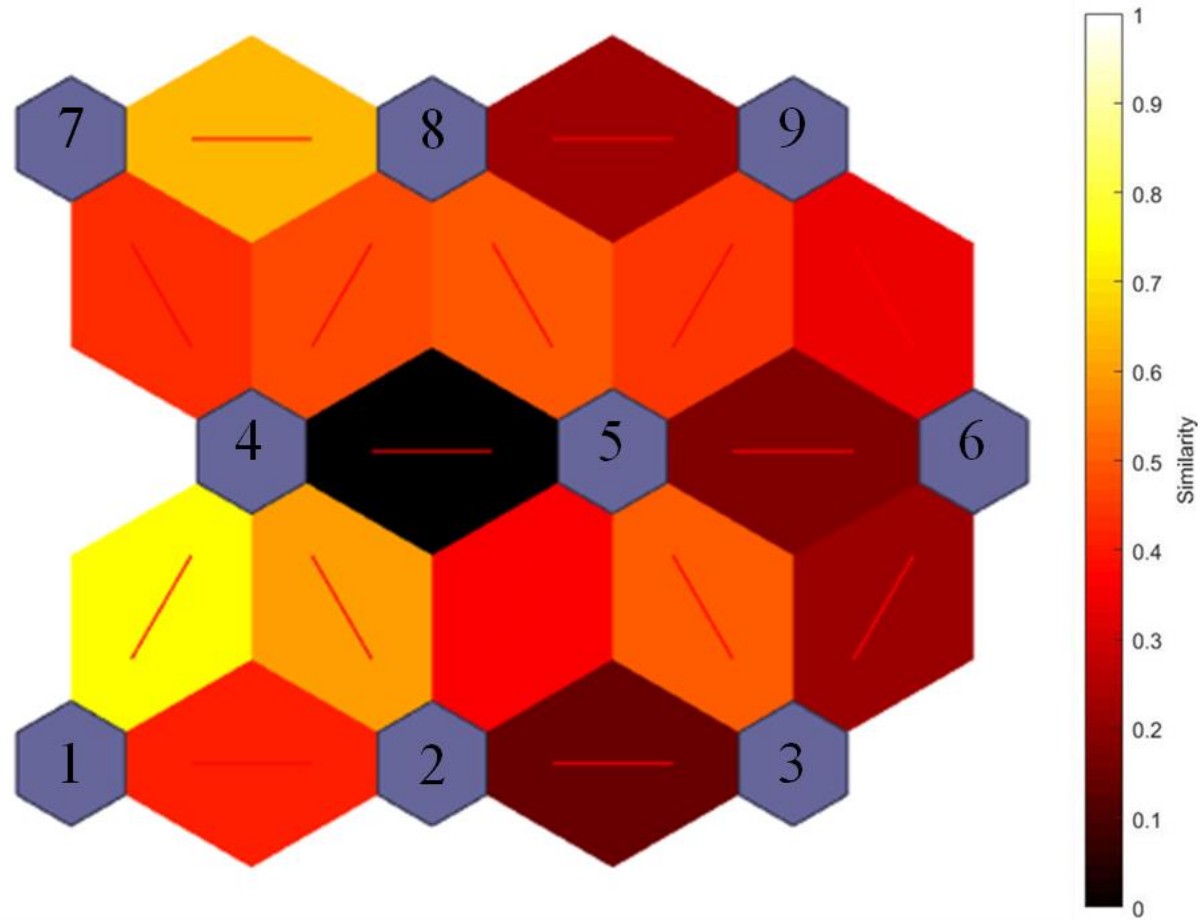

**Figure A1: Self-organizing map configuration with neighbor similarity, illustrating the differences between neighboring**
**clusters. Cluster numbers are noted in the purple nodes, note their different order compared to other Figures in the paper.**

**Appendix B: Three-dimensional PV structure**

We explore the 3D structure of the upper and lower isobaric PV. This structure of the PV sheds light on the tropopause height and the low-level PV signature, highlighting the existence of "PV towers", known to occur during the mature phase of the most intense cyclones (Rossa et al., 2000, Čampa and Wernli 2012). A PV tower is characterized by values greater than 0.5 PVU throughout the atmospheric column, representing a column vortex extending from the surface to the tropopause. Here, a simplified 3D PV structure is shown by the lowermost pressure level of the 2-PVU surface for upper-level PV and the uppermost level of the 0.5-PVU surface for low-level PV (Figs. B1 and B2). Specifically, the PV field was scanned for values greater than 0.5 PVU across the 800 hPa level, as below the 800 hPa level the pressure surfaces may intersect the mountains, leading to unrealistically large PV anomalies. The minimum pressure where the PV drops below 0.5 PVU is defined as the upper boundary of the low-level PV surface for a given cyclone. This method successfully filters out topographic influences and emphasizes the PV signature of the surface cyclone.

One surprising fact arising from the present study is that the stormiest MCs are attributed to AWB cluster 2, despite AWB being expected to lead to a rapid decay of the cyclone. Moreover, surface cyclones attributed to this cluster are not the deepest in terms of SLP (not shown). Instead, the surface cyclones seem contracted, resulting in sharper SLP gradients. The results suggest that the cyclones deepen due to the AWB streamers' southward extension and the cyclonic tilting at their tip, just above the surface cyclone. This setup induces extreme PV anomalies across the tropospheric column and leads to the formation of prominent tropospheric PV towers. Note that the PV towers are obscured by the averaging, thus do not cross the tropopause. Nevertheless, the tallest low-level PV features are evidence of a larger likelihood for the presence of PV towers that enter the stratosphere. As shown in Fig. B1, cluster 2 (and 8, to a lesser extent) extends farthest from the main PV reservoir (lowered tropopause along the northern part of the domain) and breaks cyclonically on lower levels. The response in mid-level is shown in Fig. B2, in the form of the tallest low-level PV tower coupled to the lowermost dip of the tropopause (and vice-versa, e.g., clusters 6 and 9). This explains both the precipitation and surface-wind maxima of clusters 2 and 8, compared to the other clusters in the corresponding seasons.

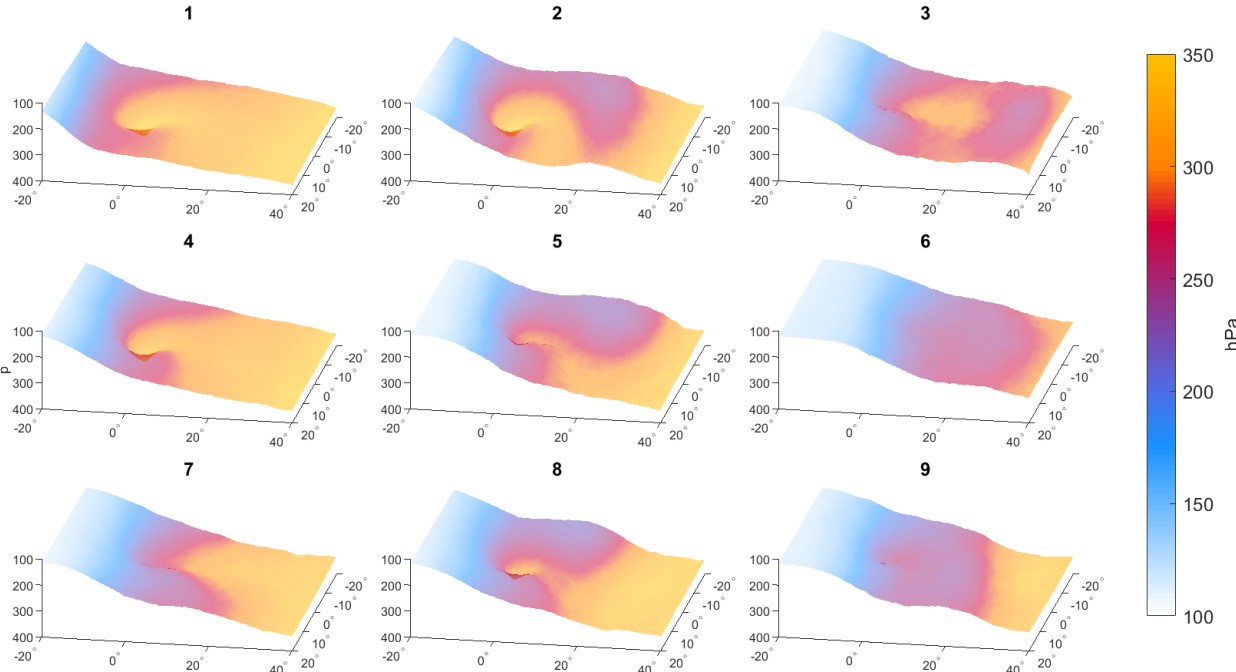

**Figure B1: Cluster composite of the lower boundary of the 2 PVU surfaces, at classification time (minimum SLP) in pressure coordinates (hPa), as viewed from the northeast.**

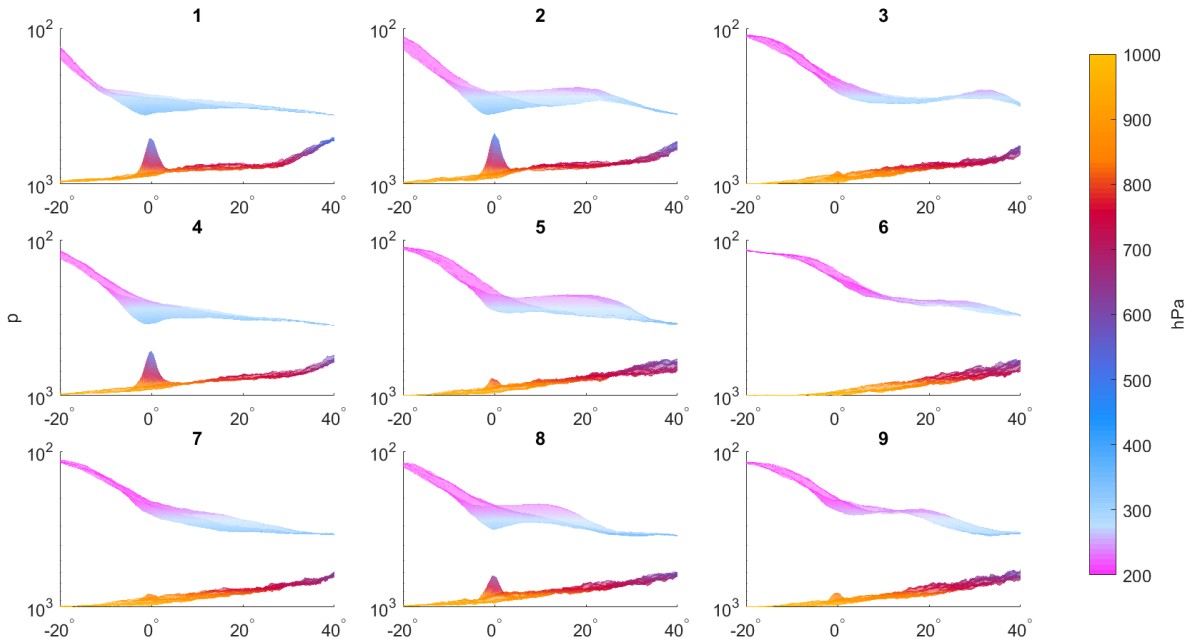

**Figure B2: As in B1, showing the y-z plane across the cyclone center, coupled to the upper boundary of the .5 PVU surface, illustrating the presence of PV towers.**

**Appendix C: PV streamers and cut-offs, WCB, and fronts**

The presence of PV streamers varies strongly between clusters (Fig. C1), with the peak frequencies of both PV streamers and cut-offs lying mostly along the equatorward border of the trough. For some clusters, the average PV feature itself is detected as a PV streamer (clusters 2, 5, and 8), and for others (clusters 3, 4, 6, and 7), the streamers are smoothed in the PV composite, an indication of their smaller scale and large spatial variability. Though possibly on lower isentropic levels, PV cut-offs often overlap with the streamers and are significantly less frequent for winter clusters. The exception is cluster 7 (daughter cyclones) where the cut-off is distinctly separated from the streamer. This suggests that daughter cyclones form at the tip of a PV streamer emanating from the synoptic system, possibly forming a cut-off as they approach their maximum intensity (i.e., classification time). PV cut-offs are most dominant in clusters 6 and 9, and again while the cut-off is evident in the PV composite for cluster 9, it is not evident for cluster 6. This suggests weaker and/or more scattered PV cut-offs (but most frequent) in cluster 6.

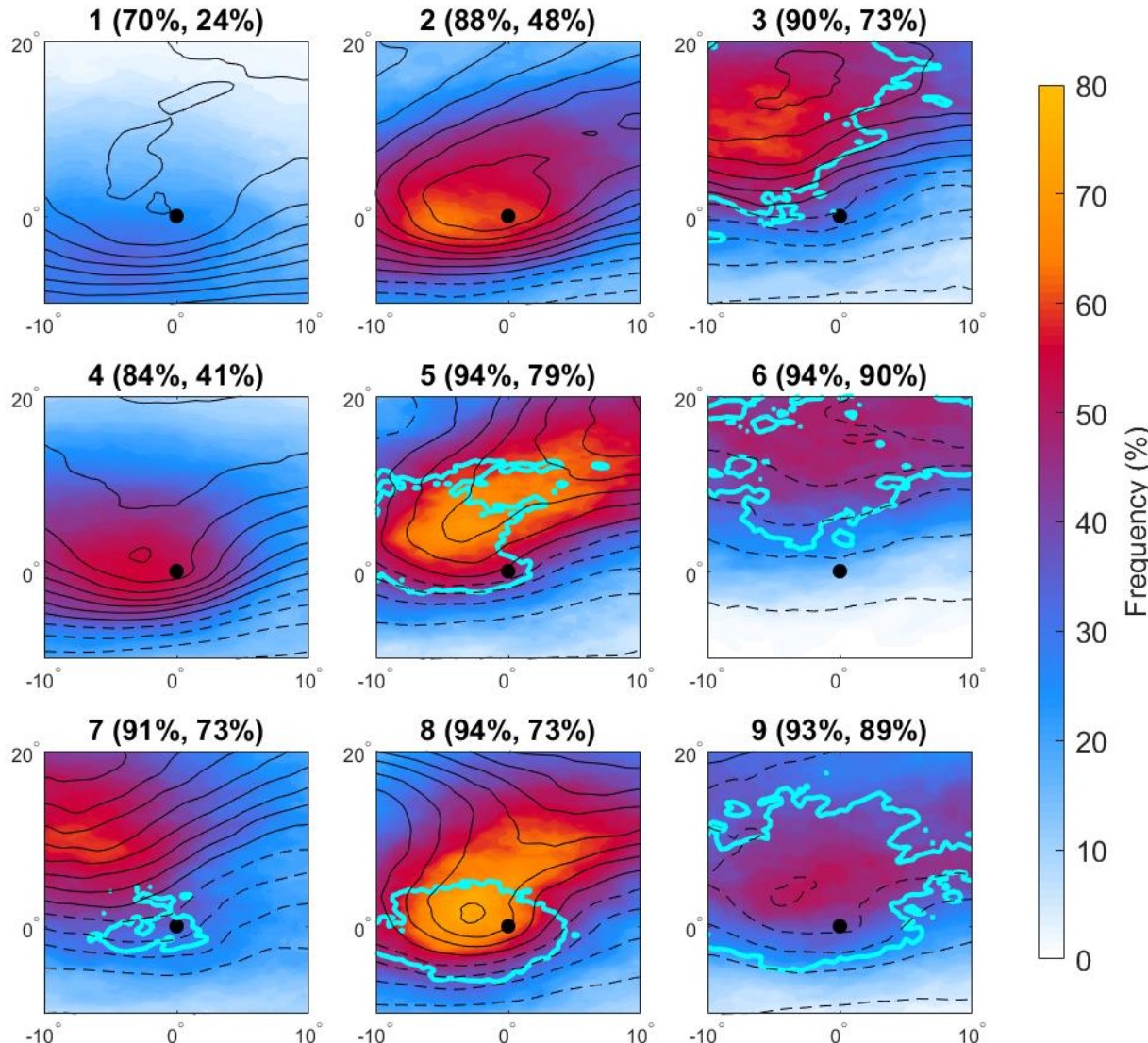

**Figure C1: Composites of PV streamers frequency (%, shading) and 10% contour for PV cut-off frequency (cyan), integrated over the 320-340 K isentropes, in 5 K intervals. A streamer/cut-off mask is applied to grid points where a streamer/cut-off is identified on at least one isentropic level. Cluster frequencies of PV streamers and cut-offs, respectively, are noted in the title, where the streamer frequency accounts for any signal within the domain, and cut-off frequency accounts for cyclones that report a cut-off present in at least 5 grid points. Black contours show cluster PV composites (as in Fig. 3, 0.5 PV intervals), with dashed contours for values below 3 PVU. Note the smaller domain compared to Fig. 3.**

Mid-tropospheric WCB reveals the uneven contribution of diabatic effects between the different cyclones clusters (Fig C2). While a WCB is reported across 90% of most clusters, cluster 6 is distinguished with a mere 50% frequency, as expected from the dry environment where it is found. The frequency of WCB peaks in cluster 4, possibly due to the advection of moist air as the lee-cyclone propagates over the Mediterranean, transitioning from stage A (cluster 1) to stage B (cluster 4). This conclusion is supported by a tongue of warm equivalent potential temperatures on the 850 hPa surface noted for cluster 4 but much weaker for cluster 1 (Fig. C3).

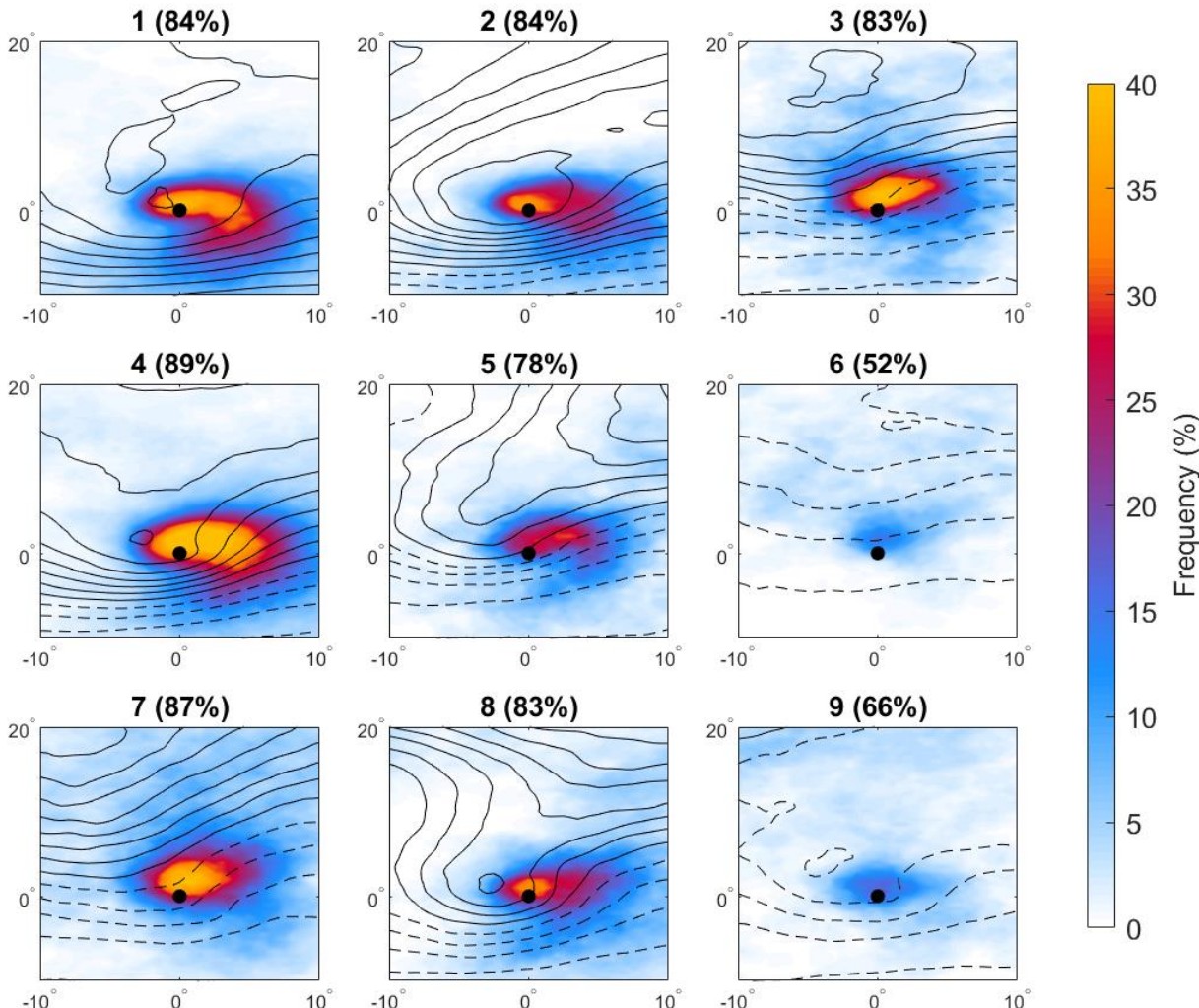

**Figure C2: As Fig. C1 but with shading for mid-troposphere (800-400 hPa) warm conveyor belt (WCB) masks frequency. The cluster-mean frequency where a WCB is detected somewhere in the domain is reported in the panel titles.**

Cluster composites of equivalent potential temperatures ($\theta_e$, Fig. C3) cover a wide range of temperatures. Besides the different extremes (clusters 1 and 6), the patterns vary from meridional gradients (clusters 3, 5, and 8) to partly zonal ones (clusters 2 and 4), to near homogeneous conditions (clusters 1, 6, and 9). Clusters 2, 4, and 8 display occluding air as a warm tongue curving into the cyclone center from the south. The large difference between clusters 1 and 4, given their similar geography and surface temperature anomalies, appears to indicate the dominant role of moisture in type B cyclones, compared to type A.

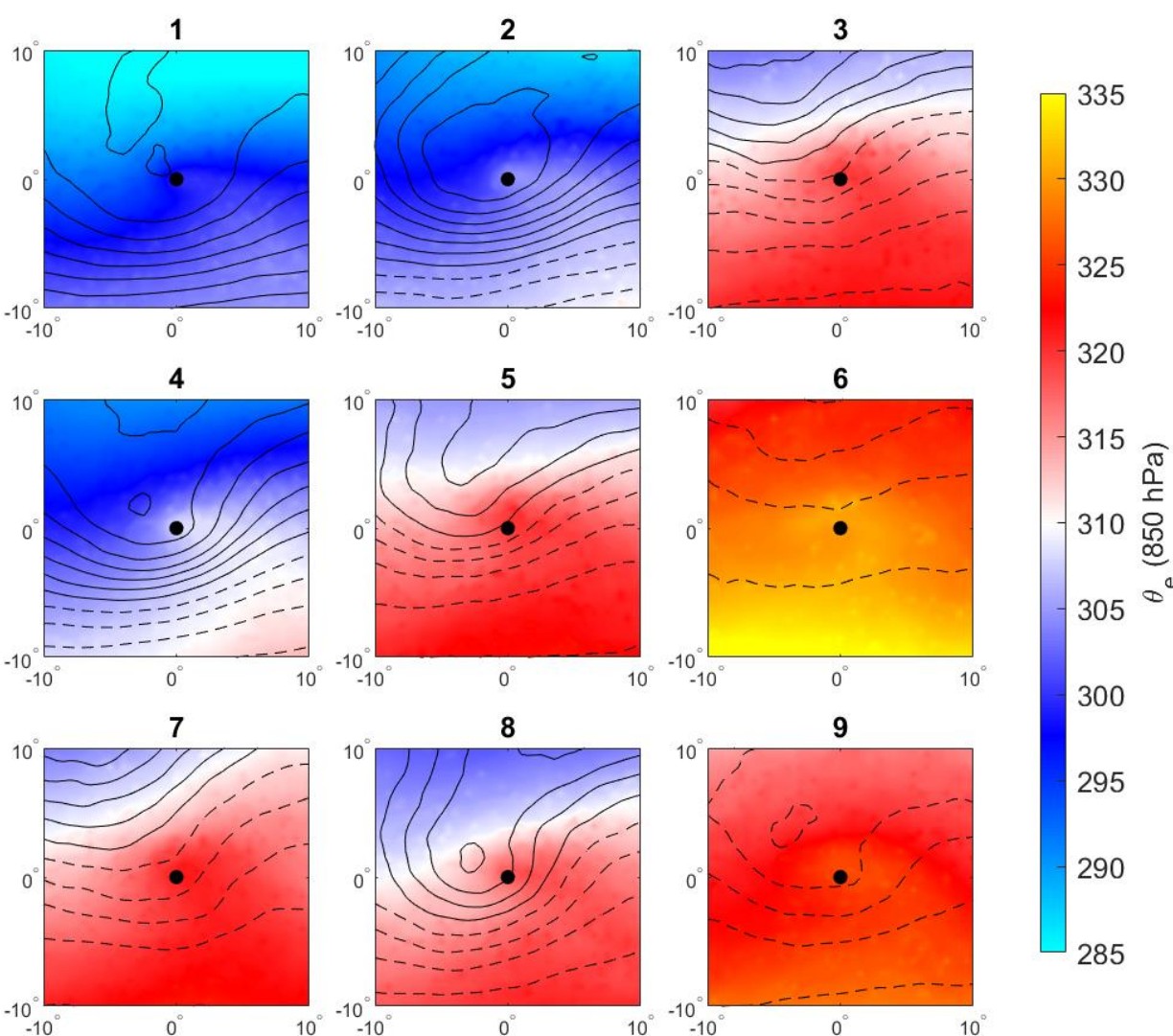

**Figure C3: As in C1 but with shading of 850-hPa $\theta_e$ (K).**

Meteorological fronts are an objectively challenging feature to handle with means of composites, as they are identified as long, thin objects at varying locations relative to the cyclone. Nevertheless, front mask composites (Fig. C4) show warm and cold fronts emanating east and west of the center, respectively, for at least 50% of the cyclones. A trailing cold front is evident for clusters 2, 4, and 8. The warm fronts are relatively spatially consistent throughout the clusters, though the averaged frequencies differ from 36% for warm fronts in cluster 6 to 81% for cluster 1.

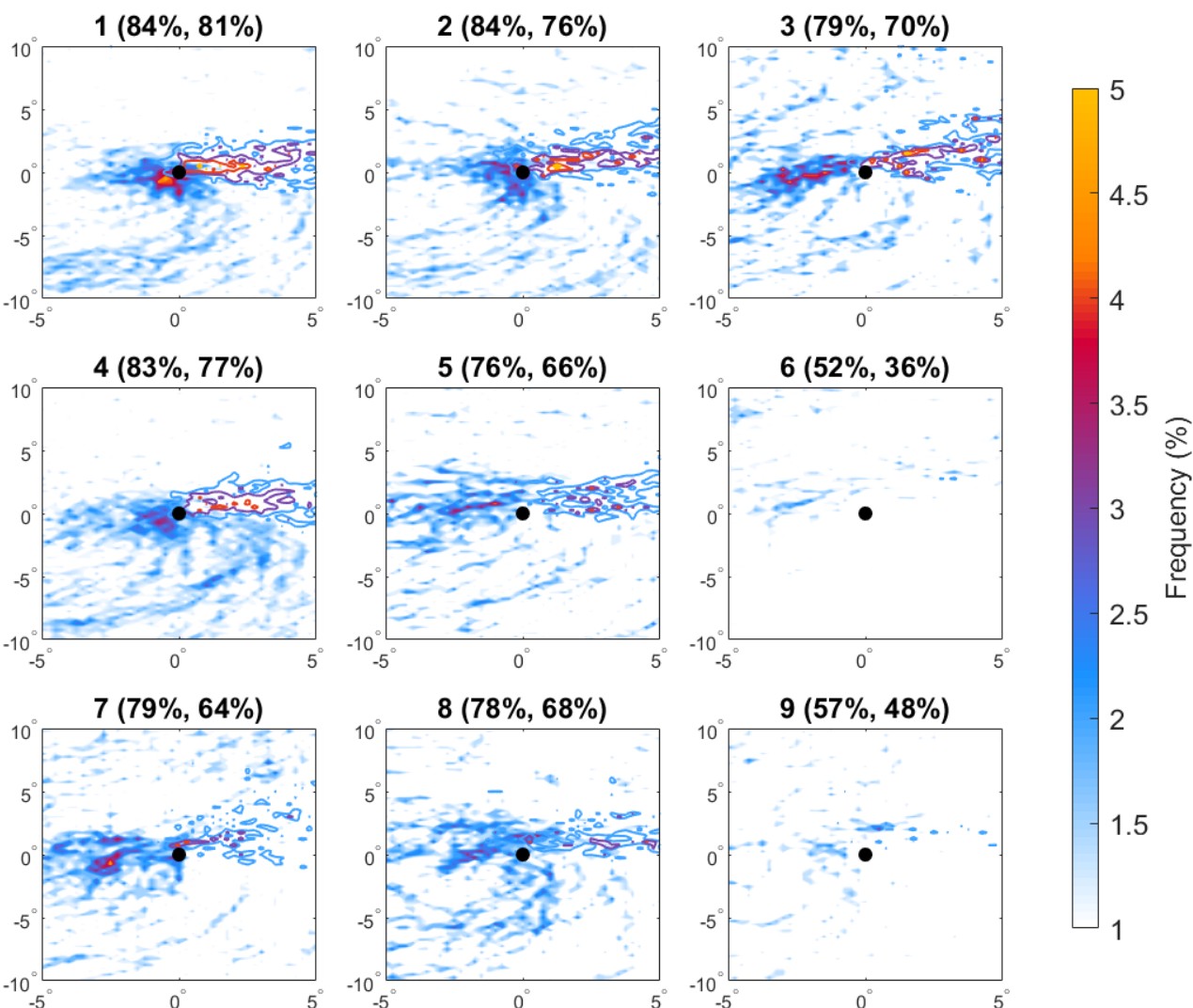

**Figure C4: Composite cluster frequencies of cold fronts (shading) and warm fronts (contours). The cluster-mean frequencies of a nonzero front signal are reported in the panel titles (left-cold, right-warm), corresponding to the fraction of cyclones showing a front present somewhere in a $10^0$ box around the cyclone.**


**Appendix D: Distribution of Medicanes between clusters**

On the continuum between tropical and extra-tropical cyclones, Medicanes are a specific set of tropical-like cyclones forming in the Mediterranean (extra-tropical environment). This elusive class of MCs was first noted in the 1980s (see Table 1). However, until now the features that render an MC as a Medicane were mostly subjective, relying on cloud structure observations and the presence of a warm core. To date, an objective, consistent definition for Medicane is still pending.

Attempting yet to put Medicanes into the context of the PV-based classification here, we examine a list of 12 agreed Medicanes from the literature (Table D1). The table consists of all Medicanes for which a warm core has been confirmed (personal communication, see acknowledgement section). While too few cases are considered to make conclusive statements, it still appears that most winter Medicanes are captured as a stage B lee cyclone and AWB+CWB cyclones, while autumn cases are mostly captured as short-wave cut-off lows (clusters 4, 2, and 9

respectively, 25% of the Medicanes each). Recalling that the first (cluster 4) is in line with the characterization of cluster 4 cyclones as "free traveling waves" as opposed to the stationary cluster 1, these Medicanes are more likely to attain their warm core initially through warm seclusion (see Fig. C3). The latter (cluster 9), given the extended warm anomalies, possibly induce intense ocean evaporation and diabatic heat release to grow as Medicanes. Indeed, both mechanisms are considered among the drivers of different Medicanes. As for cluster 2 Medicanes, these include the

category B case shown in Miglietta and Rotunno (2019), where the Medicane maintains its connection with the main PV reservoir. However, since the MC clusters are defined based on the minimum SLP time of the cyclone, and tropical characteristics may develop in different life stages of the cyclone, the association of PV clusters to Medicanes cannot, by construction, explain the full lifecycle development of the Medicanes. Moreover, the spatial scale in which Medicanes are usually diagnosed is much smaller than the large-scale structures examined here. Thus, Medicanes do

not form under a single large-scale PV scenario, and a systematic investigation of the PV distribution of Medicanes is yet an open issue.

| Dates | Cluster | Name | Area | References |
|---|---|---|---|---|
| 19820124-28 | 2 | Leucosia | S Med | Ernst & Matson (1983); Pytharoulis et al. (2000), Reed et al. (2001) |
| 19830927-1002 | 9 | Callisto | W Med | Pytharoulis et al. (2000) |
| 19950114-18 | 2 | Celeno | E Med | Cavicchia & von Storch (2012); Pytharoulis et al. (1999); Pytharoulis et al. (2000) |
| 19961006-11 | 5 | Cornelia | C Med | Cavicchia & von Storch (2012); Pytharoulis et al. (2000); Miglietta and Rotunno (2019) |
| 20051213-16 | 2 | Zeo | E Med | Fita et al. (2007); Luque et al. (2007); Miglietta et al. (2013); Fita and Flaounas (2018); Dafis et al. (2000); Miglietta and Rotunno (2019); Miglietta et al. (2021) |
| 20060925-26 | 9 | Maria/Querida | C Med | Moscatello et al. (2008); Davolio et al. (2009); Miglietta et al. (2011); Chaboureau et al. (2012); Miglietta et al. (2013); Miglietta et al. (2015); Dafis et al. (2020) |
| 20111104-09 | 4 | Rolf | W Med | Miglietta et al. (2013); Dafis et al. (2018); Dafis et al. (2020) |
| 20140119-22 | 4 | ilona | W Med | Cioni et al. (2016); Dafis et al. (2020) |
| 20141107-09 | 5 | qendresa | S Med | Homar et al. (2003); Carrió et al. (2017); Cioni et al. (2018); Pytharoulis et al. (2018); Mylonas et al. (2019); Noyelle et al. (2019); Bouin et al. (2020); Dafis et al.(2020) |
| 20171116-19 | 4 | Numa | Ionian Sea | Marra et al. (2019); Dafis et al. (2020) |
| 20180927-30 | 9 | Zorbas | Ionian Sea | Dafis et al. (2020); Kouroutzoglou et al. (2021) |
| 20200915-18 | 6 | Udine-Cassilda-Ianos | S Med | Lagouvardos et al. (2021); Prat et al. (2021); D'Adderio et al. (2022); Zimbo et al. (2022) |

*Table D1: Date (YYYYMMDD), cluster number, name, and references and of 12 confirmed Medicane cases.*

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
