# Peer review of "Process-based classification of Mediterranean cyclones using potential vorticity"

_EGUsphere, 2023_

## Author Response (AR1)

We appreciate and thank both reviewers for their insightful comments, which enhance the scope and readability of the manuscript. In the following we respond to each of the reviewers' comments point by point, with the reviewers' comments in blue, and our response in black.

Reviewer #1

**This study deals with the clustering of Mediterranean cyclones based on dynamic classification (upper-level PV). I think the results are relevant for the entire scientific community working on cyclones in the Mediterranean, but also outside the area, considering that the methodology can be applied anywhere. In my opinion, two results deserve particular attention: the observation of different trends for different clusters, which were only marginally visible in historical data until now; the fact that the cluster category remains the same since cyclogenesis, thus providing additional predictability. Therefore, my suggestion is for minor revision, mainly focused on improving the readability of the article and its structure.**

We thank the reviewer for this valuable input. We agree that the method is relevant regardless of geographic location, and we intend to evaluate its performance in other regions and globally in the future.

We did our best to adhere to and implement the useful comments.

**MINOR POINTS:**

**L93: I think it is not appropriate to consider tropical-like cyclones as thermal lows. They are also not scattered in the Mediterranean (they are mainly concentrated near the Balearic Islands and the Ionian Sea).**

Thank you for the opportunity to clarify. The sentence is changed to "heat lows scattered across the Mediterranean" to avoid referring to tropical-like cyclones. We now address Medicanes more clearly in Appendix D adhering to reviewer #2.

**L98: explosive development can also occur for extratropical cyclones, therefore the mixing of information with medicanes is not appropriate in this case; therefore, I would keep the two sentences separate.**

These sentences were separated.

**L110: Portmann et al. (2020)**

corrected.

**L113: I cannot find Buzzy et al. (2022) among the references.**

corrected to Buzzy et al. (2020).

**L177: The reason for choosing 9 clusters is not clearly addressed even in the Appendix.**

Thank you for this comment, the issue is now clarified in Appendix A:

"The number of clusters (N=9) is selected using the elbow method (Liu and Deng, 2020), as the number beyond which a sharp decrease in the SOM learning rate (i.e., SOM error reduction per added cluster) is evident".

For clarity, Fig. R1 shows the SOM errors as a function of number of clusters tested.

[Figure]

*Figure R1: SOM MSE (member deviation from cluster composite, averaged over all clusters) for different number of clusters. Note that the reduction of MSE between 9 and 12 clusters is smaller than between 6-9, representing a slower learning rate above 9 clusters.*

**L200: I suggest moving these details about statistical methods to an Appendix.**

We see the point of moving some details to an appendix, however, due to the brevity of subsection 2.4, we find the present structure easier to follow.

**L219: "contour points with small great circle distance (anchor points)": please clarify what you mean, adding some additional information to better explain the methodology.**

The text was changed to the following:

"PV streamers are also identified on single isentropic levels. To this aim, on each isentropic level, the 2-PVU contour delimiting the stratospheric PV reservoir is extracted, and then PV streamers are identified based on this contour as narrow, elongated disturbances. More specifically, for each point along the contour it is checked whether there exists a corresponding further point along the contour such that (i) the great-circle (baseline) distance between these two anchor points is less than 2000 km, and (ii) the along-contour distance between the two points exceeds the baseline distance by at least a factor 2. A more detailed description of the algorithm, including some examples, can be found in the Supplement of Sprenger et al. (2017)".

**L234: therefore, the fronts are identified along the inflections of the gradient ( )? Should not it be along the maxima of the gradient ( )? Please explain.**

As nicely illustrated in Figure (3) of Hewson (1998), fronts are placed on the warm side of the strongest gradients of the considered thermal variable. A reference to the figure was added to the main text.

**L260: How do you explain the difference between large-scale and convective precipitation distribution? Should not it be the convective precipitation more localized along the cold front?**

Thanks for highlighting this point. Indeed, convective precipitation is more localized, but not only along the cold front. There is usually a secondary maximum near the cyclone center, which is enhanced by the composite, due to the cyclone-centered perspective (Raveh-Rubin and Wernli, 2016).

**L280: I suggest naming the different clusters only in Section 4, after analyzing the different properties.**

We appreciate this suggestion. While we agree that the naming of the clusters is more easily understood after the discussion of the results, for some of the clusters, the main patterns are best illustrated by the PV itself. As it facilitates the reading to address clusters by name rather than by number, we prefer to use the cluster names throughout the discussion of the results.

**L280: Are cyclonically (8) and anticyclonically (2, 5) breaking PV streamers named from the analysis of the similarities to the theoretical configuration given in Thorncroft's study?**

Yes, the different life cycles (AWB, CWB as well as cutoffs) are concluded based on the visual patterns as suggested by the composites, and their similarity to what is expected from literature. SLP patterns also reinforce the separation of the different baroclinic life cycles. We want to note that the separation is conceptual and that these lifecycles may be detected in other clusters as well, but not as frequently nor as distinctly as in the corresponding clusters (i.e., cluster 5 for AWB, 8 for CWB, and 2 for both).

**L281: I suggest turning to "The frequency of objectively identified PV streamers".**

changed as suggested.

**L290 (L430): what does S1 (S3) mean?**

changed to Appendices A1 and B2.

**L368: To overcome these interpretation issues, could you also look at the 12-hour interval intensification rates?**

Thank you for the opportunity to address this matter. We realize that the negative Bergeron numbers may raise questions. Figure R2 shows the deepening rate with the Bergeron number computed for 12 hours. The problem with 12-hour deepening rates is that the values are much larger than the 24-hour criterion, as the deepening phase considered is shorter. Therefore, it is difficult to compare the two. We chose to show the 24-hour window as it is more commonly used in literature. Furthermore, the negative Bergeron numbers are useful to highlight cases with relatively early peak times, as discussed in the main text. We also compute the deepening rates 12 and 24 hours prior to classification time (minimum SLP), still, it is clear from Fig, R3 that some cyclones peak at even earlier stages, primarily clusters 1 and 4 that are linked to lee-cyclogenesis, which is characterized by a rapid initial deepening.

[Figure]

*Figure R2: same as Fig. 7 in the main article but using a 12-hour sliding window instead of 24-hour.*

[Figure]

*Figure R3: Bergeron numbers based on the 12 (red) and 24 (blue) hours preceding minimum SLP time. Percentages correspond to the fraction of cyclones captured in each division. i.e., 19% (43%) of the cyclones included in the analysis peak in less than 12 (24) hours from their genesis.*

**L424: "weakest for clusters 6 and 9": it does not seem to me that they are the weakest in terms of wind anomaly, they are the weakest in terms of PV anomaly.**

We thank you for making this comment. We now realize that using a significance filter on the shaded field was somewhat misleading. The wind-speed anomalies were only shown when significant at the 99% confidence level with respect to gridded monthly climatology, applying rather strict criteria. Clusters 6 and 9 are indeed showing significantly weaker anomalies, weaker than the non-significant anomalies of clusters 5 and 8, which are closer to the average of all considered MCs. Considering both reviewer's inputs, we moved the figure to section 3.2 while removing the filter for the 10-m wind anomalies, and instead, we now show deviations of absolute 10-meter wind speed from the cyclone mean.

**L427: "anticyclonic breaking PV streamers": this creates some confusion since you classify Cluster 8 as "CWB low" in Table 1.**

We changed this phrase to "initially anticyclonically breaking PV streamers". As shown in the cyclogenesis PV composites, both clusters 2 and 8 initiate with an AWB event, followed by cyclonic wrapping of the PV streamer.

**L535: most instead of more.**

changed as suggested.

**L538: windward side instead of wind side?**

changed as suggested.

**L455: "downward movement into the lower levels" is more appropriate here than downdrafts.**

changed as suggested.

**L451-L471: the narration here does not work smoothly, since some statements (L458-459; L463: "Anti-cyclonic followed by cyclonic wave breaking") refer to points that will be better discussed in the following Section 4 and in Appendix B (Fig. B1), but they are not adequately supported here.** Thank you for this notion. We now reference figure B1. Still, both the anticyclonic and the cyclonic portion of the PV streamers are evident in both the PV composites of cluster 2 and 8. The cyclonic part is indeed better illustrated on isobaric PV, as in Fig. B1, but we feel that this point is important to explain the maximum PV anomalies obtained for these two clusters.

**L463: a new paragraph should start from "The presence of…".**

changed as suggested.

**L535: most instead of more.**

changed as suggested.

**L538: windward side instead of wind side?**

changed as suggested.

**L607: Appendix B: Considering this part is central to the discussion, I ask the authors to consider moving it into the main text.**

We appreciate the suggestion and acknowledge that this change may facilitate the understanding of some of the results. However, this part of the analysis is not very intuitive, and we feel that such a change may eventually disrupt the flow of the paper. Thus, we prefer to leave the 3D analysis as an appendix. Yet, we now reference this section earlier on, in section 3.2.

**L635: Figure B2: No PV towers can be detected here: is this due to the coarse resolution of the reanalyses?**

Thank you for this clarification. Indeed, the composited "PV towers" also reflect the frequency of individual members, and do not extend into the stratosphere. This is due to many individual members that have weaker low-level PV signals, leading to a reduction in the height of the average PV tower. However, "complete" PV towers are most frequent for clusters 2 and 8. To avoid manipulation of the mean patterns, we kept the figure as is, and added a clarification to this point.

Reviewer #2

**This study classifies nine types of cyclones in the Mediterranean by applying the Self Organizing Map analysis to the upper-level potential vorticity fields around cyclones. In spite of a simple classification based on a single variable, it effectively identifies different types of cyclones including lee-cyclones, Rossby wave breakings, and heat lows with different characteristics such as three-dimensional structures and seasonality. These results are nicely summarized in Table 1. I agree that the process-based classification transcends the geographical-seasonal classification because a similar process can occur in different regions and different processes can occur in the same region. This study further examines the predictability and long-term trends. Overall, this study clearly demonstrates the variability of Mediterranean cyclones with appropriate, sophisticated, and straightforward analyses. While the manuscript is almost acceptable for the publication in Weather and Climate Dynamics in the current form, I would like to make three suggestions (major comments 1-3) which may improve the manuscript. Therefore, I recommend minor revisions of this manuscript at this stage.**

We thank the reviewer for these comments and suggestions that indeed improve the readability of the manuscript. We did our best to address the issues raised.

**Major comments**

**In section 4, is it possible to discuss whether individual classes are similar to or different from cyclone types in other regions? Although the authors discuss it for some classes (e.g., a description of clusters 2 and 8 in Lines 557-558), it would be helpful to summarize such discussion after Table 1. For example, are some clusters in the present study similar to clusters for the cyclones in the Southwest Pacific (Catto 2018,** https://doi.org/10.1175/JCLI-D-17-0746.1**)? Readers may also want to know whether some classes are linked to medicanes because the authors explain it in detail in section 1.**

Thank you, we appreciate these suggestions. We added a sentence regarding the similarity to previous cyclone classifications. However, we find it difficult to compare Mediterranean cyclones to cyclones in other regions. First, their horizontal scale is usually smaller compared to oceanic extra-tropical cyclones. Second, the modification of cyclones by topography is substantial (Flaounas et al., 2022), and leads to greater variability. Third, the influence of heat-lows is more dominant in the Mediterranean. We feel a more in-depth comparison is beyond the extent of the present manuscript. This is certainly an objective we intend to address and will be easier to handle once we extend the proposed PV classification to global data.

As for Medicanes, the lack of a formally accepted definition for such events makes a thorough analysis of their distribution between clusters challenging. Nevertheless, we now directly address Medicanes in Appendix D featuring an agreed-upon set of Medicanes.

**The authors should provide more information on precipitation analysis. Regarding the large-scale and convective precipitation, are they outputs from a global model used for the ERA-5 reanalysis? Regarding the warm conveyor belt analysis, did the authors apply the trajectory analysis introduced by Madonna et al. (2014) to the hourly wind fields of the ERA-5 reanalysis? In particular, I wonder to what extent the relevant results are changed if the analyses are applied to a higher-resolution dataset that can resolve convection more explicitly. While I think these analyses in the present manuscript are still informative (the authors do not need to remove them), the authors should discuss caveats on these analyses. Another minor**

**suggestion is that the authors show the sum of large-scale and convective precipitation by shading or contour in Fig. 2d.**

We have added the following text to section 2.4: "Both large-scale and convective precipitation are provided by the ERA-5 reanalysis product, using output from parametrization schemes (Hersbach et al., 2020). While these results are parametrized and are expected to show sensitivity to model resolution (especially convective precipitation), it does allow drawing conclusions regarding the sign and relative response of precipitation under the different PV classes".

We have further clarified points concerning WCBs in subsection 2.5.2 and changed the shading of Fig. 2d as suggested.

**Lines 420-434: This paragraph describes 10-m wind speed and upper-level PV. I wonder why the upper-level PV is discussed in section 3.4 which focuses on surface impact. I think that the analysis of the upper-level PV anomaly is more closely linked to Figs. 3 and C1 which are explained in section 3.2. On the other hand, this paragraph focuses less on 10-m wind speed. From the perspective of surface impact, I think that the total wind speed is also important. Fig. 2d and 10 only show the anomaly concerning monthly local climatology. Does it make sense to examine the total wind speed?**

Thank you for this suggestion. Indeed, the total wind speed was also examined, and we agree it is also relevant to understand the surface response. We therefore added the following figure to section 3.4 and moved the PV and wind speed anomaly plot to section 3.2, removing the significance filter which was misleading.

**Minor comments**

**4.  Lines 422-426: The wind anomaly for cluster 5 looks weaker than that for clusters 6 and 9.**

See response to reviewer #1

**5.  Line 445: Give the cluster numbers for the summer clusters.**

Added

**6.  Lines 463- 465: The explanation of warm conveyor belt and Fig. C3 should be moved before the sentence that refer to Fig. C2 first in Lines 456-457.**

Changed as suggested.

**7.  Lines 544-545: Give the section of the Appendix.**

Added.

**8.  Fig. B2: is this the y-z plane through the cyclone center?**

Indeed, it is. The caption was edited for clarity.

---

## Author Response (AR2)

We thank the reviewers for the additional suggestions. We did our best to address all comments. We here respond to each point raised by the Reviewer 2 (reviewer comments in blue, with our response in black).

Reviewer #2

**I appreciate the authors' efforts to revise the manuscript based on the reviewers' comments. In particular, Appendix D helps readers understand the relationship between the PV-based classification and medicanes. I would like the authors to think about the following minor points before the publication of this paper.**

Thank you, we agree that extending the discussion to include Medicanes does add value to the manuscript.

**1. Lines 501-502: What do the authors mean by cyclonic and anticyclonic tilts?**

Thank you for the opportunity to rephrase this term. To avoid confusion with vertical tilt, we now use the term "curvature" to describe the direction in which the PV features rotate. E.g., the PV feature shown in the composite of cluster 2 (Fig. 3 in the main text) shows anticyclonic curvature in its northeastern part, and cyclonic curvature closer to the cyclone center.

**2. Fig. B1: If possible, denote which direction is the north in the figure, or replace "(viewed from the) northeast" with "east-northeast" for accuracy.**

Thank you for this remark, the axes orientation was added for clarity and the captions were changed as suggested.

**3. Lines 738-740: This sentence should be rephrased without using "respectively" to link the three cluster numbers with the relevant cluster characteristics more clearly.**

The sentence was split into the following two sentences:

*"While too few cases are considered to make conclusive statements, it appears that most winter Medicanes are captured as a stage B lee cyclone (cluster 4) and AWB+CWB cyclones (cluster 2, 25% of the Medicanes each). Autumn cases are mostly captured as short-wave cut-off lows (cluster 9) that form another 25% of the Medicanes."*